# ATXN3 regulates lysosome regeneration after damage by targeting K48-K63-branched ubiquitin chains

Maike Reinders [ID][1,6], Bojana Kravic[1,6], Pinki Gahlot [ID][1], Sandra Koska[1], Johannes van den Boom [ID][1], Nina Schulze [ID][2], Sophie Levantovsky [ID][3], Stefan Kleine[4], Markus Kaiser[4], Yogesh Kulathu [ID][5], Christian Behrends [ID][3] & Hemmo Meyer [ID][1✉]

## Abstract

The cellular response to lysosomal damage involves fine-tuned mechanisms of membrane repair, lysosome regeneration and lysophagy, but how these different processes are coordinated is unclear. Here we show in human cells that the deubiquitinating enzyme ATXN3 helps restore integrity of the lysosomal system after damage by targeting K48-K63-branched ubiquitin chains on regenerating lysosomes. We find that ATXN3 is required for lysophagic flux after lysosomal damage but is not involved in the initial phagophore formation on terminally damaged lysosomes. Instead, ATXN3 is recruited to a distinct subset of lysosomes that are decorated with phosphatidylinositol-(4,5)-bisphosphate and that are not yet fully reacidified. There, ATXN3, along with its partner VCP/p97, targets and turns over K48-K63-branched ubiquitin conjugates. ATXN3 thus facilitates degradation of a fraction of LAMP2 via microautophagy to regenerate the lysosomal membrane and to thereby reestablish degradative capacity needed also for completion of lysophagy. Our findings identify a key role of ATXN3 in restoring lysosomal function after lysosomal membrane damage and uncover K48-K63-branched ubiquitin chain-regulated regeneration as a critical element of the lysosomal damage stress response.

**Keywords** Lysosome; Stress Response; Ubiquitin; Membrane; Autophagy
**Subject Categories** Autophagy & Cell Death; Organelles; Post-translational Modifications & Proteolysis

## Introduction

Lysosomal membrane permeabilization (LMP) or full rupture of the limiting membrane of lysosomes and late endosomes constitutes severe cellular stress relevant in various conditions such as neurodegeneration, infection, and cancer. Several conditions cause lysosomal damage, including exposure to lysosomotropic compounds, lipid peroxidation, and unbalanced lipid compositions associated with neurodegeneration, aging, or cancer, as well as cellular uptake of silica, crystals, or pathogens. Cells have developed a complex response, termed the endo-lysosomal damage response (ELDR), that consists of distinct branches (Meyer and Kravic, 2024; Yang and Tan, 2023; Zoncu and Perera, 2022). Lysosomes with minor damage undergo quick repair of the limiting membrane and re-acidification, which is mediated by the ESCRT machinery and by replenishment of lipids at damage-induced ER-lysosome contact sites (Herbst et al, 2020; Radulovic et al, 2018; Radulovic et al, 2022; Skowyra et al, 2018; Tan and Finkel, 2022). If repair fails, individual terminally damaged lysosomes are triaged for destruction by a form of selective macroautophagy, termed lysophagy (Hoyer et al, 2022; Hung et al, 2013; Maejima et al, 2013). In parallel, an mTOR-governed signaling pathway induces the biogenesis of new lysosomal components (Jia et al, 2018). In addition, however, a number of recent reports describe, for a subset of lysosomes, damage-induced tubulation and transport events, which are regulated by various factors such as LRRK2, RAB7, or conjugation of ATG8 to single membranes (Bhattacharya et al, 2023; Bonet-Ponce et al, 2020; Cross et al, 2023). Conversely, invagination mediated by microautophagy has been shown to eliminate sections of the limiting membrane and selective degradation of individual lysosomal membrane proteins (Lee et al, 2020; Ogura et al, 2023). This suggests that these processes contribute to the regeneration of functional lysosomes following the initial membrane sealing efforts.

A key regulatory feature of the endo-lysosomal damage response is the extensive ubiquitylation of damaged lysosomes (Eapen et al, 2021; Kravic et al, 2022; Maejima et al, 2013) by a number of

[1]Molecular Biology I, Center of Medical Biotechnology, Faculty of Biology, University of Duisburg-Essen, Essen, Germany. [2]Imaging Center Campus Essen, Center of Medical Biotechnology, Faculty of Biology, University of Duisburg-Essen, Essen, Germany. [3]Munich Cluster for Systems Neurology, Medical Faculty, Ludwig-Maximilians-Universität München, Munich, Germany. [4]Chemical Biology, Center of Medical Biotechnology, Faculty of Biology, University of Duisburg-Essen, Essen, Germany. [5]MRC Protein Phosphorylation and Ubiquitylation Unit, University of Dundee, Dundee, UK. [6]These authors contributed equally: Maike Reinders, Bojana Kravic. ✉E-mail: hemmo.meyer@uni-due.de

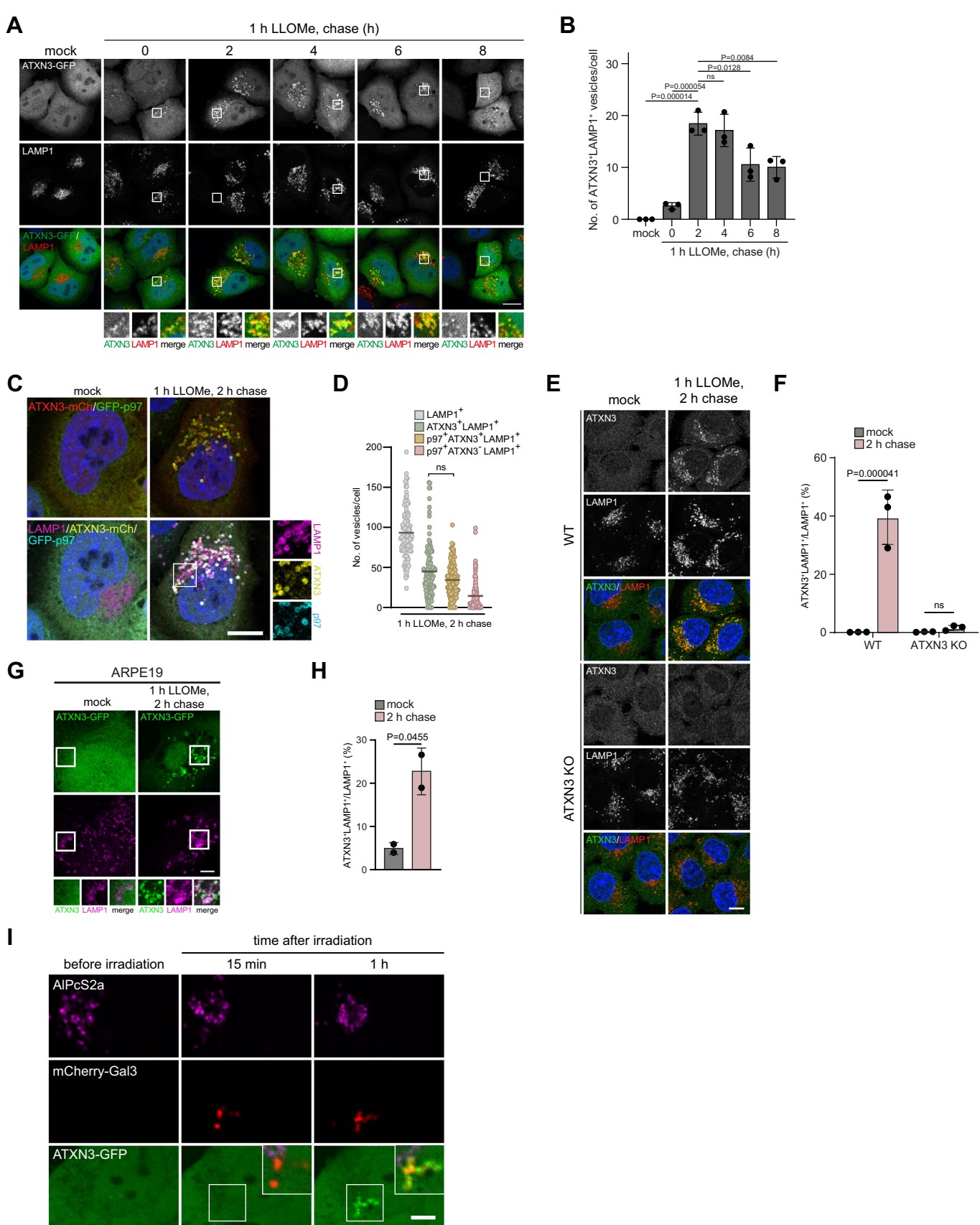

**Figure 1. ATXN3 translocates to LAMP1-positive compartments at a late stage after lysosomal damage.**

(A) HeLa cells expressing ATXN3-GFP were mock-treated or LLOMe-treated (1 mM) for 1 h, fixed at indicated time points after LLOMe washout, and stained with LAMP1 antibodies. Note that ATXN3 translocation peaks at 2 h after washout. Scale bar, 15 µm. (B) Quantification of (A). $n = 3$ biologically independent experiments with >40 cells per condition. One-way ANOVA with Tukey's multiple comparison test (mock vs. 0 h chase $P = 0.7088$; 2 h chase $P < 0.0001$; 4 h chase $P < 0.0001$; 6 h chase $P = 0.0016$; 8 h chase $P = 0.0024$). The graph shows mean ± S.D. (C) HeLa cells expressing ATXN3-mCherry and the GFP-p97 (E578Q) substrate-trapping mutant were treated and stained as indicated. Note the robust colocalization on a subset of LAMP1-positive compartments. Scale bar, 10 µm. (D) Quantification of (C). $n = 3$ biological replicates with >30 cells per condition per experiment. One-way ANOVA with Tukey's multiple comparison test. The line indicates the mean. (E) Immuno-detection of endogenous ATXN3 on damaged lysosomes. ATXN3 KO cells served as control. Scale bar, 10 µm. (F) Quantification of (E), $n = 3$ biological replicates with >22 cells per condition per replica. Two-way ANOVA with Sidak's multiple comparisons test. Error bars represent mean ± S.D. (G) Neuronal ARPE19 cells transiently expressing ATXN3-GFP were mock or LLOMe-treated (1 mM) for 1 h, followed by 2 h chase before fixation and LAMP1 staining. Scale bar, 5 µm. (H) Quantification of (G). $n = 2$ biologically independent experiments with >23 cells per condition per replica. Unpaired two-tailed $t$-test. Error bars represent mean ± S.D. (I) Light-induced lipid peroxidation. Lysosomes in HeLa cells expressing ATXN3-GFP and the damage marker mCherry-Gal3 were loaded with photosensitizer AlPcS2a. Cells were pulse-irradiated and imaged live at the indicated time points. Affected lysosomes are bleached and decorated by Gal3. Note the recruitment of ATXN3-GFP only after 1 h. Scale bar, 5 µm. Source data are available online for this figure.

ubiquitin ligases identified so far (Chauhan et al, 2016; Gahlot et al, 2024; Liu et al, 2020; Teranishi et al, 2022; Yoshida et al, 2017). An initial wave of ubiquitylation that includes K63-linked ubiquitin chains serves to recruit autophagy receptors and triggers phagophore formation for lysophagy (Eapen et al, 2021; Gahlot et al, 2024). However, ubiquitylation regulates additional steps beyond constituting an "eat-me" signal for lysophagy. These functions include the removal of the actin stabilizer CNN2 by the ubiquitin-directed AAA-ATPase VCP/p97 to further promote phagophore formation (Kravic et al, 2022). Intriguingly, ubiquitylation of lysosomal compartments continues to increase even after phagophore formation around some lysosomes and peaks around 3 h after damage initiation, and this coincides with a further increase in VCP/p97 recruitment (Papadopoulos et al, 2017). The role of this late ubiquitylation is as yet unclear, suggesting an unknown function beyond repair and lysophagy in restoring lysosomal functionality.

Using a proteomics approach, we previously identified Ataxin-3 (ATXN3) as a potential regulator in the endo-lysosomal damage response (Koerver et al, 2019). ATXN3 is a deubiquitinating enzyme that cooperates with VCP/p97 in various pathways, including promoting autophagic flux (Ashkenazi et al, 2017; Pfeiffer et al, 2017; Wang et al, 2006). Mutation of ATXN3 is associated with spinocerebellar ataxia type 3, a hereditary neurodegenerative disease (McLoughlin et al, 2020). Here, we show that ATXN3 translocates to lysosomes upon damage and targets conjugates with K48-K63-branched ubiquitin chains, such as LAMP2 to facilitate degradation in the lysosome, likely through microautophagy in order to promote lysosome regeneration and to thus restore functionality of the lysosomal system.

## Results

### ATXN3 translocates to lysosomes in response to lysosomal membrane permeabilization

We first asked whether ATXN3 localizes to lysosomes upon lysosomal damage. To permeabilize lysosomes, we initially applied L-leucyl-L-leucine methyl ester (LLOMe). LLOMe is lysosomotropic and condenses to membranolytic poly-leucine peptides specifically in late endosomes and lysosomes (Thiele and Lipsky, 1990), thus recapitulating damage in pathophysiological conditions. ATXN3-

GFP distributed diffusely in unchallenged HeLa cells but robustly translocated to LAMP1-positive compartments upon LLOMe treatment (Fig. 1A,B). The translocation occurred relatively late in the damage response, with ATXN3 emerging at the end of the 1 h LLOMe treatment and peaking 2 h after LLOMe washout (Fig. 1A,B). ATXN3 colocalized with its partner, the AAA-ATPase VCP/p97 (Fig. 1C,D), which is an established regulator of the lysosomal damage response (Eapen et al, 2021; Klickstein et al, 2024; Papadopoulos et al, 2017). We confirmed translocation of endogenous ATXN3 to damaged lysosomes with a specific antibody (Fig. 1E,F). Moreover, we showed that ATXN3-GFP translocates to damaged lysosomes also in retina pigment epithelium ARPE19 cells (Fig. 1G,H), a non-transformed neuronal cell model (Calcagni et al, 2023). Furthermore, spatially restricted induction of lysosomal membrane permeabilization by light-stimulated lipid peroxidation (Hung et al, 2013) led to translocation of ATXN3-GFP to affected lysosomes (Fig. 1I). Thus, ATXN3 is recruited to lysosomes damaged in various ways to different degrees and in different cell types suggesting that ATXN3 is a general element of the lysosomal damage response.

### ATXN3 is essential for restoring lysosomal degradative capacity after damage

Damaged lysosomes become decorated with cytosolic galectins, including galectin-3 (LGALS3 and Gal3) that bind to exposed luminal glycans and are easily detectable by immunostaining (Jia et al, 2018; Maejima et al, 2013; Thurston et al, 2012). The progress of restoring lysosomal integrity during the damage response can be monitored by following the clearance of lysosome-associated Gal3 (Maejima et al, 2013). Gal3 clearance occurs initially in those lysosomes that are being repaired and later by lysophagy of a subpopulation of lysosomes that are terminally damaged, as well as by lysosome regeneration (Bhattacharya et al, 2023; Eapen et al, 2021). We observed a robust dependence of Gal3 clearance on ATXN3, and this was demonstrated by inactivating ATXN3 in three independent ways. We first generated a series of HeLa ATXN3 knockout (KO) cells that showed delayed Gal3 clearance after LLOMe-induced damage (Fig. EV1A–C). The observed delay in Gal3 clearance was rescued by overexpression of wild-type ATXN3, but not of a catalytically inactive ATXN3-C14A mutant, demonstrating that ATXN3 function in the lysosomal damage response required its deubiquitinating activity (Figs. 2A,B and EV1D). Overexpression of ATXN3 with mutation of the p97/VCP binding

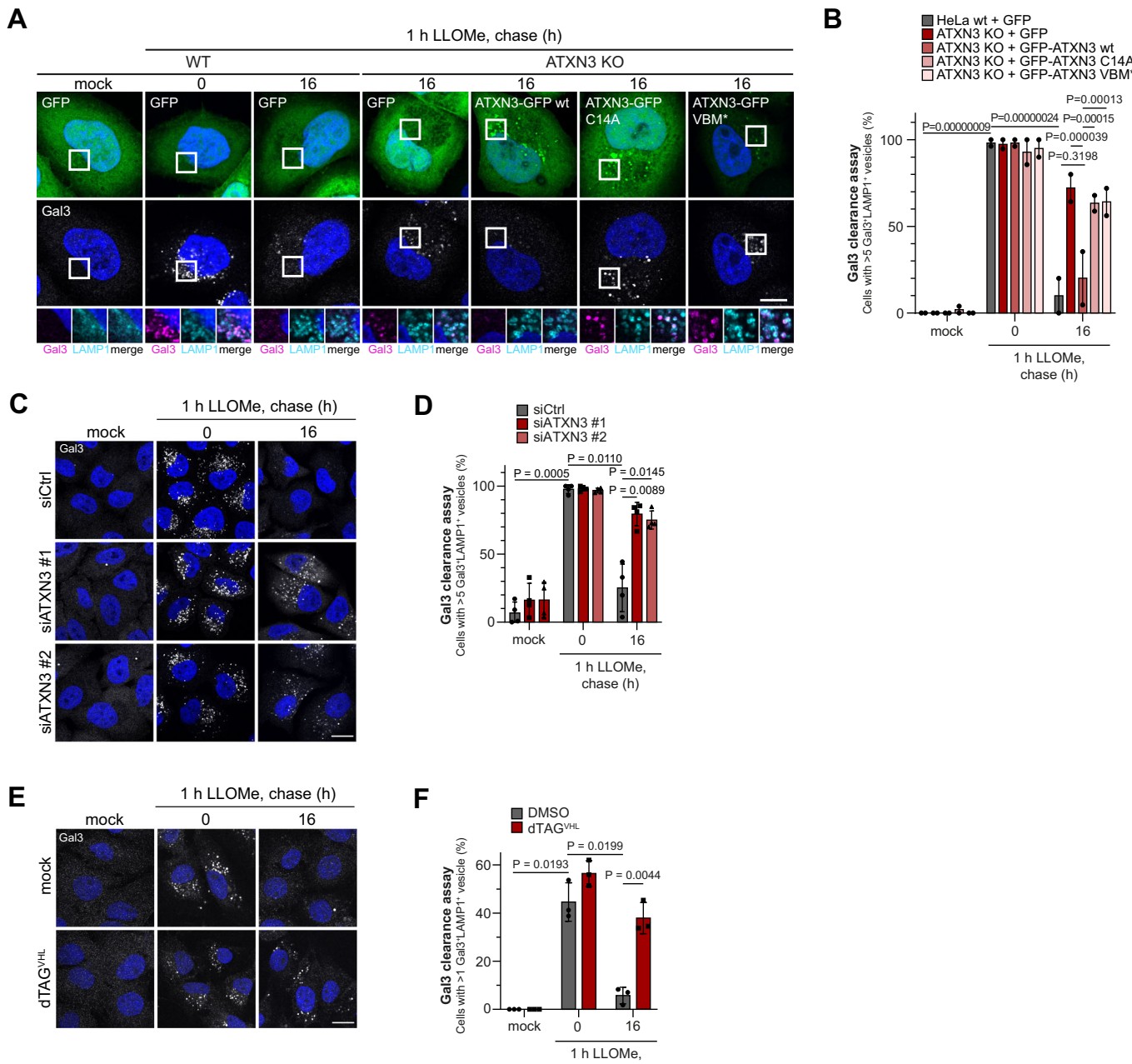

**Figure 2. ATXN3 is essential for the restoration of degradative compartments after lysosome damage.**

(A) Knockout-rescue assays for Gal3 clearance. HeLa ATXN3 KO and parental cells expressing indicated constructs were mock or LLOMe-treated (1 mM) for 1 h before washout. Cells were fixed at the indicated time points, and Gal3-positive lysosomes were immuno-stained. Note that Gal3-decorated lysosomes persisted in ATXN3 KO cells, which was rescued by re-expression of ATXN3 WT but not of the catalytically inactive ATXN3-C14A or the p97-binding deficient VBM mutant (VBM*). Scale bar, 5 µm. (B) Quantification of (A). $n = 2$ biologically independent experiments with >15 cells quantified per condition per experiment. Error bars, S.E.M. Two-way ANOVA with Tukey's multiple comparison test was used to test significance. (C) HeLa cells were treated with indicated siRNAs, and Gal3 clearance was monitored after lysosomal damage was assayed as in (A). Scale bar, 15 µm. (D) Quantification of (C). $n = 4$ biologically independent experiments with >30 cells quantified per condition per experiment. Two-way ANOVA with Tukey's multiple comparison test was used to test significance. The graph shows mean ± S.D. (E) Induced degradation of ATXN3 compromises clearance of damaged lysosomes. The ATXN3 gene was tagged with the FKBP12$^{F36V}$ tag in U2OS cells. ATXN3 degradation was induced by dTAG$^{VHL}$ treatment, and Gal3 clearance was assessed. Scale bar, 15 µm. See Fig. EV1I for degradation verification. (F) Quantification of (E), $n = 3$ biological replicates with >30 cells per condition per experiment. One-way ANOVA with Tukey's multiple comparison test. The graph shows mean ± SD. Source data are available online for this figure.

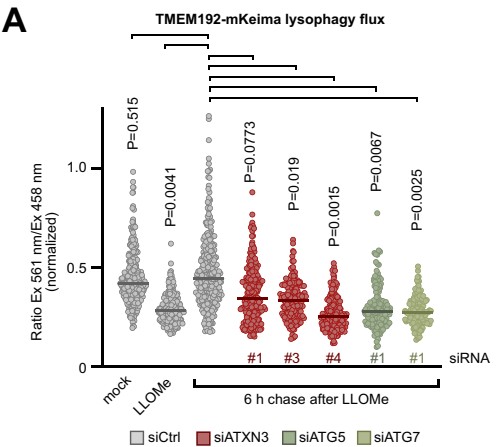

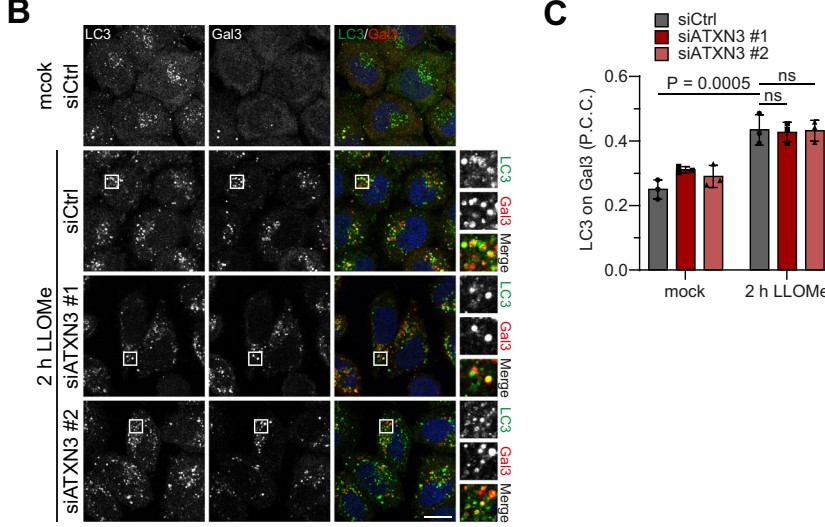

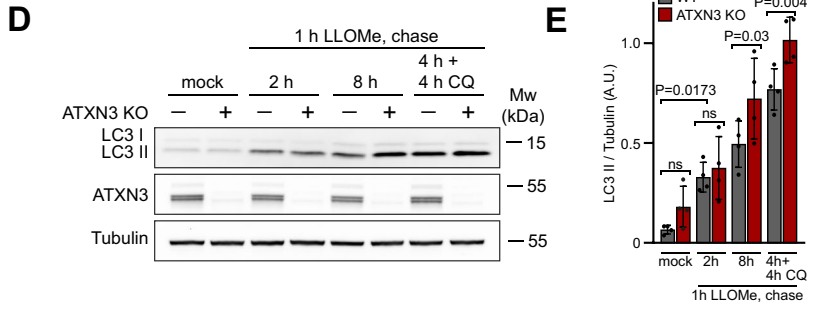

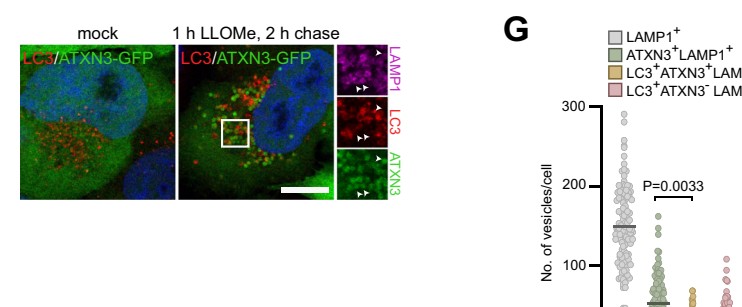

**Figure 3.   ATXN3 is required for completion of lysophagy, but not for phagophore formation.**

(A) Lysophagy assay. HeLa cells stably expressing pH-sensitive mKeima fused to the cytosolic terminus of TMEM192. Cells were transfected with indicated siRNAs and treated with LLOMe for 1 h, followed by a 6 h washout. mKeima fluorescence was assessed at the indicated excitation wavelengths, and the ratio between both intensities was determined. Note that the increase in the 561 nm/458 nm ratio is reduced in ATXN3-, ATG5-, or ATG7-depleted cells, indicating a defect in autolysosome formation. $n = 4$ biological replicates with >30 cells per condition per experiment. One-way ANOVA with Holm–Sidak's multiple comparison test. The line indicates the median. (B) Phagophore formation around damaged lysosomes is not affected by ATXN3 depletion. HeLa cells were treated with indicated siRNAs, incubated with LLOMe before fixation as indicated, and immuno-stained for LC3 and the damage marker Gal3. Note that samples were also costained for p62 (shown in Fig. EV3B). Scale bar, 15 µm. (C) Quantification of (B). $n = 3$ biologically independent experiments with >70 cells quantified per condition per experiment. Two-way ANOVA with Tukey's multiple comparison test. The graph shows mean ± SD. (D) Lysophagy flux is compromised in ATNX3 KO cells. Western blot of indicated cell lines and time points. Note that initial LC3 lipidation is not affected by ATXN3 KO at 2 h chase, but the lipidated LC3 form accumulates in ATXN3 KO after 8 h chase. (E) Quantification of (D). $n = 4$ biologically independent experiments. Two-way ANOVA with Tukey's multiple comparison test. Error bars represent the mean with SD. (F) Lack of colocalization of ATXN3 with LC3. HeLa cells expressing ATXN3-GFP were mock or LLOMe-treated as indicated, fixed, and stained for LC3. Arrowheads indicate ATXN3-positive and LC3-negative lysosomes. Scale bar, 10 µm. (G) Quantification of (F). $n = 3$ biological replicates with >30 cells per condition per experiment. One-way ANOVA with Tukey's multiple comparison test. The line indicates the mean. Source data are available online for this figure.

motif (VBM*) in ATXN3 KO cells also failed to rescue Gal3 clearance, indicating that ATXN3 needs to cooperate with p97 (Figs. 2A,B and EV1D,E). Likewise, depletion of ATXN3 in HeLa cells by two different siRNAs affected Gal3 clearance (Figs. 2C,D and EV1F). To exclude that the observed effect was indirect due to long-term depletion of ATXN3, we applied a rapid induced-degradation approach. ATXN3 was genomically tagged with FKBP12$^{F36V}$ in U2OS cells (Fig. EV1G,H). Addition of dTAG$^{VHL}$, a proteolysis targeting chimera compound linking FKBP12$^{F36V}$ with the VHL ubiquitin ligase (Nabet et al, 2020), induced degradation of ATXN3 efficiently within 1 h (Fig. EV1I). Like in the other approaches, induced degradation of ATXN3 significantly compromised Gal3 clearance after damage infliction (Fig. 2E,F). Concurring with the other approaches, this demonstrates that ATXN3 is essential for restoring the lysosomal system and its degradative capacity following damage to lysosomal membranes.

## ATXN3 is needed for the completion of lysophagy, but not for phagophore formation

In line with the late recruitment of ATXN3, we did not observe prominent colocalization of ATXN3 with the ESCRT repair factor IST1, nor a significant delay of the initial membrane repair in ATXN3 KO cells based on LysoTracker recovery (Fig. EV2A–D), nor an effect in IST1 or ALIX recruitment or release (Fig. EV2E–H). Likewise, we did not detect any effect on very early LC3 or TECPR1 recruitment that could have indicated compromised unconventional lipidation to a single membrane (Fig. EV2I–L). We therefore next asked whether ATXN3 was instead involved in lysophagy. To specifically monitor lysophagy, we used a previously established assay that is based on the pH-sensitive mKeima fused to the cytosolic tail of TMEM192 in stable HeLa cell lines (Gahlot et al, 2024; Shima et al, 2023). Due to its cytosolic location, mKeima reports on acidification in autolysosomes, which enclose the whole lysosome. In contrast, TMEM192-mKeima does not report on acidification of the lysosomal lumen during repair and thus differentiates between the pathways. Six hours after release from LLOMe treatment, control-depleted cells showed a shift from green to red-excitable mKeima, indicating a decrease in pH consistent with lysophagy occurring (Figs. 3A and  EV3A). In contrast, depletion of ATXN3 largely dampened the increase in red-excitable mKeima, as did depletion of ATG5 or ATG7 as positive controls, indicating a direct function of ATXN3 in lysophagy (Figs. 3A and

EV3A). Based on this finding, we speculated that ATXN3 may regulate the recruitment of autophagy receptors or phagophore formation. However, depletion of ATXN3 did not affect recruitment of SQSTM1/p62 (Fig. EV3B,C) or LC3 (Fig. 3B,C) to damaged lysosomes compared to control-depleted cells. Consistent with that, western blot analysis revealed that LC3 lipidation was not reduced in ATXN3 KO cells (Fig. 3D,E). Instead, lipidated LC3 accumulated in ATXN3 KO cells at later stages compared to parental cells, and this was not further increased by chloroquine treatment, showing that lysophagic flux was compromised in ATXN3 KO cells (Fig. 3D,E). In contrast, this was not observed for KO of another p97-associated DUB, YOD1, as a control (Fig. EV3D,E). Of note, PIK3C3/VPS34 and ULK1 inhibition largely reduced LC3 lipidation upon LLOMe treatment, confirming that a significant fraction of LC3 lipidation stemmed from phagophore formation and that this fraction was not affected by ATXN3 KO (Fig. EV3F,G). Consistent with that, also recruitment of ULK1 to damaged lysosomes was not compromised in ATXN3 KO cells (Fig. EV3H,I) and ATXN3-GFP only partially colocalized with LC3 (Fig. 3F,G), suggesting that ATXN3 is not involved in phagophore formation.

## ATXN3 acts on regenerating lysosomes to restore functionality of the lysosomal system

Our results so far indicate that ATXN3 is critical for the restoration of the lysosomal capacity but is not directly involved in the formation of phagophores around terminally damaged lysosomes. We therefore explored the localization of ATXN3 in more detail. SIM live-cell microscopy revealed that the majority of ATXN3-decorated LAMP1 compartments were LysoTracker-negative, indicating that they were not yet acidified within 3 h since damage infliction (Fig. 4A,B). We therefore speculated that ATXN3 localizes to a class of LAMP1 compartments that are subjected to the slower process of regeneration rather than being quickly repaired or autophagocytosed (Bhattacharya et al, 2023; Bonet-Ponce et al, 2020). Phosphatidylinositol (4,5)-bisphosphate (PI(4,5)$P_2$) has been implicated as a marker for regenerating lysosomes (Bhattacharya et al, 2023). We used the pleckstrin homology (PH) domain of PLCD1 fused to GFP as an established PI(4,5)$P_2$ sensor, which localized to the plasma membrane in control cells, as expected (Stauffer et al, 1998). Of note, the GFP-PH sensor translocated to ATXN3 and LAMP1-positive compartments

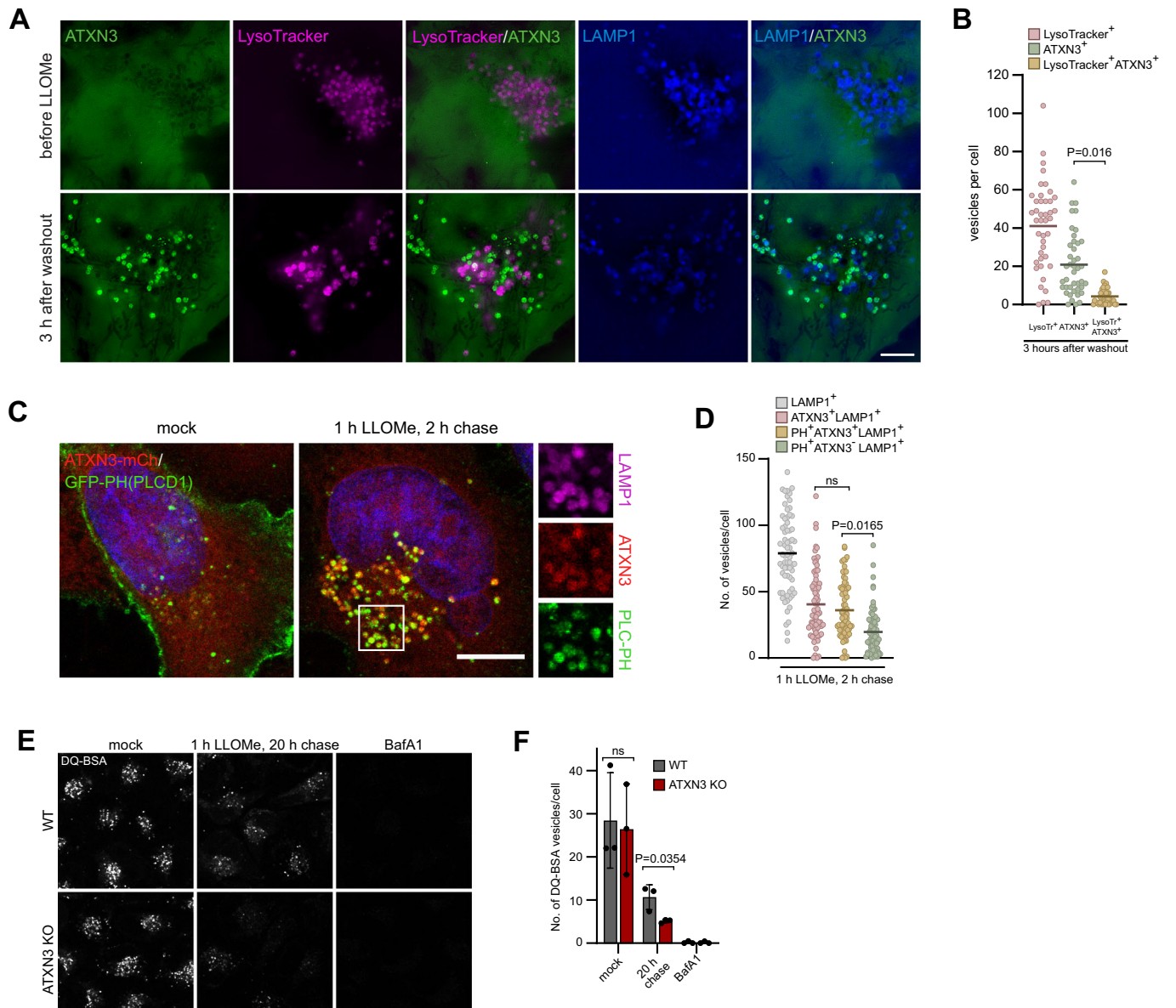

**Figure 4. ATXN3 localizes to non-acidified regenerating lysosomes.**

(A) 3D-SIM live-cell imaging of stable HeLa LAMP1-BFP cells loaded with lysotracker and expressing ATXN3-GFP. Cells were treated with 1 mM LLOMe for 12 min, washed, and followed during recovery in medium containing lysotracker for 3 h. Scale bar, 5 µm. (B) Quantification of 2D-SIM images taken in experiments shown in (A). $n = 2$ biologically independent experiments with >20 cells quantified per experiment. Lines indicate the median. (C) HeLa cells expressing ATXN3-mCherry and the PI(4,5) P2 sensor GFP-PH (PLCD1) were mock or LLOMe-treated as indicated and immuno-stained for LAMP1. Note the colocalization of ATXN3 and GFP-PH on a subpopulation of lysosomes. Scale bar, 10 µm. (D) Quantification of (C), $n = 3$ biological replicates with >20 cells per condition per experiment. One-way ANOVA with Tukey's multiple comparison test. The line indicates the mean. (E) DQ-BSA assay for proteolytic activity in lysosomes. HeLa parental and ATXN3 knockout cells were LLOMe-treated for 1 h. DQ-BSA was added 13 h after washout, and DQ-BSA fluorescence in lysosomes was imaged at a total of 20 h after LLOMe washout. Scale bar, 10 µm. (F) Quantification of (E), $n = 3$ biological replicates with >40 cells per condition per experiment. Mixed effects analysis with Tukey's multiple comparison test. Error bars represent the mean with SD. Source data are available online for this figure.

specifically after lysosomal damage induction (Fig. 4C,D). Consistent with the delay in Gal3 clearance observed above in ATXN3-deficient cells, we observed a reduced recovery of lysosomal degradative capacity after damage in ATXN3 KO cells as monitored by the fluorogenic proteolysis reporter DQ-BSA in lysosomes (Fig. 4E,F). Thus, ATXN3 primarily acts on regenerating lysosomes and thereby helps restore the functionality of the lysosomal system.

## ATXN3 and VCP/p97 target damage-induced K48-K63 branched ubiquitin conjugates on regenerating lysosomes

We next aimed to address the molecular basis for ATXN3 function in the lysosomal damage response. ATXN3 was recently found to bind and cleave ubiquitin chains with branched K48 and K63 linkages and to cooperate with VCP/p97 to process these

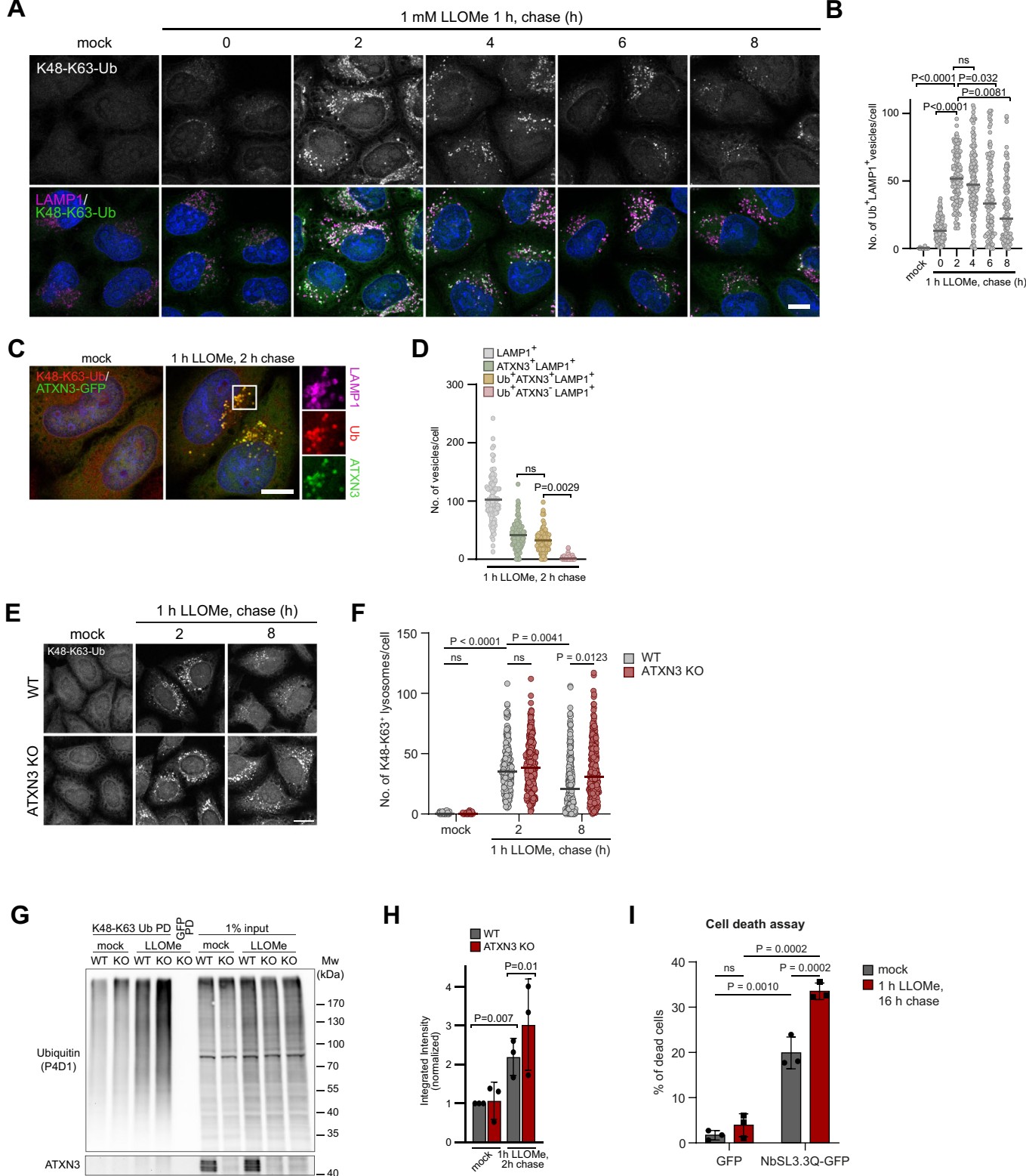

**Figure 5. ATXN3 and VCP/p97 target and turnover damage-induced K48-K63 branched ubiquitin conjugates on regenerating lysosomes.**

(A) HeLa cells were LLOMe-treated and chased for the indicated periods of time. Cells were fixed and immuno-stained with LAMP1 antibodies and an AlexaFluor568-conjugated nanobody NbSL3.3Q specific for K48-K63-branched ubiquitin chains. Note the nanobody signal peaking at 2 h chase on a subpopulation of lysosomes and fading off within 8 h. Scale bar, 10 µm. (B) Quantification of (A), $n = 3$ biological replicates with >30 cells per condition per experiment. One-way ANOVA with Holm–Sidak's multiple comparison test (mock vs. 0 h chase $P = 0.0652$; 2 h chase $P < 0.0001$; 4 h chase $P < 0.0001$; 6 h chase $P < 0.0001$; 8 h chase $P = 0.0004$). The line indicates the mean. (C) HeLa cells expressing ATXN3-GFP were fixed and stained for K48-K63-branched ubiquitin chains with NbSL3.3Q. Scale bar, 10 µm. (D) Quantification of (C), $n = 3$ biological replicates with >30 cells per condition per experiment. One-way ANOVA with Tukey's multiple comparison test. The line indicates the mean. (E) HeLa WT or ATXN3 KO cells were pulse-treated with LLOMe, chased for the indicated times, and stained with NbSL3.3Q. Note the persistence of K48-K63-branched chains in ATXN3 KO cells. Scale bar, 15 µm. (F) Quantification of (E) $n = 3$ biological replicates with >50 cells per condition per experiment. Two-way ANOVA with Tukey's multiple comparison test. The line indicates the median. (G) Damage-induced K48-K63-branched ubiquitin chains accumulate in ATXN3 KO cells. HeLa parental and ATXN3 KO cells were mock or LLOMe-treated. Branched chains were affinity-isolated from lysates with NbSL3.3Q and detected with pan-ubiquitin antibody P4D1. (H) Quantification of (G). $n = 3$ biologically independent experiments. Two-way ANOVA with Uncorrected Fisher's LSD test. Error bars represent mean with S.D. (I) Determination of dead cells by propidium iodide (PI) staining and flow cytometry analysis in GFP-expressing or NbSL3.3Q-GFP-expressing cells following LLOMe treatment. $n = 3$ biologically independent experiments with >3000 cells per condition per experiment. Two-way ANOVA with Fisher's LSD test. The graph shows mean ± SD. Source data are available online for this figure.

conjugates (Lange et al, 2024). We employed a nanobody, NbSL3.3Q, that specifically detects K48-K63-branched ubiquitin chains (Lange et al, 2024) to ask whether K48-K63 chains played any role in ELDR. Of note, overexpressed NbSL3.3Q-GFP colocalized with lysosomes specifically after lysosomal damage, indicating that K48-K63-branched ubiquitin chains constitute a signal in the lysosomal damage response (Fig. EV4A,B). To avoid interfering with the dynamics of K48-K63-branched ubiquitin chains due to NbSL3.3Q binding in overexpressing cells, we used chromophore-conjugated NbSL3.3Q for immunostaining of fixed cells. The staining revealed that K48-K63 chains peaked late during the damage response at 2 h chase and decreased over the full 8 h of analysis (Fig. 5A,B), comparable to the ATXN3 dynamics shown above. Importantly, co-imaging showed that K48-K63-branched chains and ATXN3 largely colocalized on LAMP1-positive compartments (Fig. 5C,D). Moreover, the K48-K63-branched chains also colocalized with p97 (Fig. EV4C,D) and GFP-PH (Fig. EV4E,F), suggesting that p97 and ATXN3 cooperate in targeting K48-K63 conjugates on regenerating lysosomes. Indeed, chemical inhibition of p97, knockout of ATXN3, or knockdown of ATXN3 led to an enhanced accumulation and persistence of K48-K63-branched chains as visualized by NbSL3.3Q staining, showing that ATXN3, in cooperation with p97, turns over the K48-K63-branched ubiquitin conjugates (Figs. 5E,F and EV4G,J). Affinity-isolation of K48-K63-branched chains with NbSL3.3Q biochemically confirmed LLOMe-induced formation of branched chains and accumulation in ATXN3 KO cells (Fig. 5G,H). This was specific for ATXN3, because KO of the other p97-associated DUB, YOD1, did not lead to branched chain accumulation (Fig. EV4K,L). We next expressed NbSL3.3Q to intentionally interfere with branched chain turnover (Lange et al, 2024) and asked how this affected cell survival after lysosomal damage. Notably, NbSL3.3Q expression specifically increased cell death compared to control expression after LLOMe treatment (Fig. 5I), indicating that processing of K48-K63-branched ubiquitin chains is essential for cells to cope with the stress induced by lysosomal damage.

## ATXN3 facilitates lysosomal damage-induced degradation of LAMP2

We next aimed to identify targets modified with K48-K63 branched chains in response to lysosomal damage. We treated cells with LLOMe or vehicle alone and used the NbSL3.3Q nanobody to specifically isolate conjugates with K48-K63-branched ubiquitin chains from cell lysates, as demonstrated previously (Lange et al, 2024). Comparative mass spectrometry identified a set of proteins specifically isolated from cells with damaged lysosomes (Fig. 6A and EVTab1). We focused on LAMP2 that was previously shown to be ubiquitylated and degraded in the lysosome upon lysosomal damage (Yoshida et al, 2017). Using denaturing immunoprecipitation, we confirmed an increase in ubiquitylation of LAMP2 upon LLOMe treatment, and this was further increased in ATXN3 KO cells, indicating turnover of ubiquitylated LAMP2 by ATXN3 (Fig. 6B). Detection was with a pan-ubiquitin antibody, because the K48-K63-specific nanobody does not work in Western blots. Importantly, using Western blot analysis, we confirmed damage-induced degradation of LAMP2, but less so of LAMP1, in LLOMe-treated cells (Fig. 6C,D), indicating selectivity of degradation, as previously reported (Yoshida et al, 2017). Of note, degradation of LAMP2 was delayed in ATXN3 KO cells or upon treatment with Bafilomycin A1 (Fig. 6E,F), showing that ATXN3 facilitates degradation of LAMP2 in the lysosome. Recent work demonstrated micro-autophagy as a critical element in the response to lysosome damage and as a basis for selective degradation of individual proteins in the lysosome (Lee et al, 2020). Notably, we found that knockdown of the kinase STK38, which is a prominent regulator of damage-induced microautophagy (Ogura et al, 2023) attenuated degradation of LAMP2 (Fig. EV5A,B), and that STK38 colocalized with ATXN3 and branched ubiquitin chains with significant overlap (Fig. 6G,H) suggesting that ATXN3 positively regulates microautophagy on damaged lysosomes.

## Discussion

In this study, we uncover a key role of ATXN3 in the cellular response to lysosomal membrane permeabilization. By analyzing cancer and neuronal cell models, we find that compromising ATXN3 function leads to a severe delay in clearing and regenerating damaged lysosomes and thus in restoring functionality of the lysosomal system after lysosomal damage. The fact that ATXN3 translocates to lysosomes in response to damage and that it processes stress-induced ubiquitin-conjugates on lysosomal membranes indicates that the function of ATXN3 is direct. Previous work showed that ATXN3 regulates the stability of the PI3 kinase class III complex component beclin-1/BECN1, which triggers phagophore formation, and that ATXN3 thus supports long-term autophagic flux (Ashkenazi et al, 2017; Hill et al, 2021). This

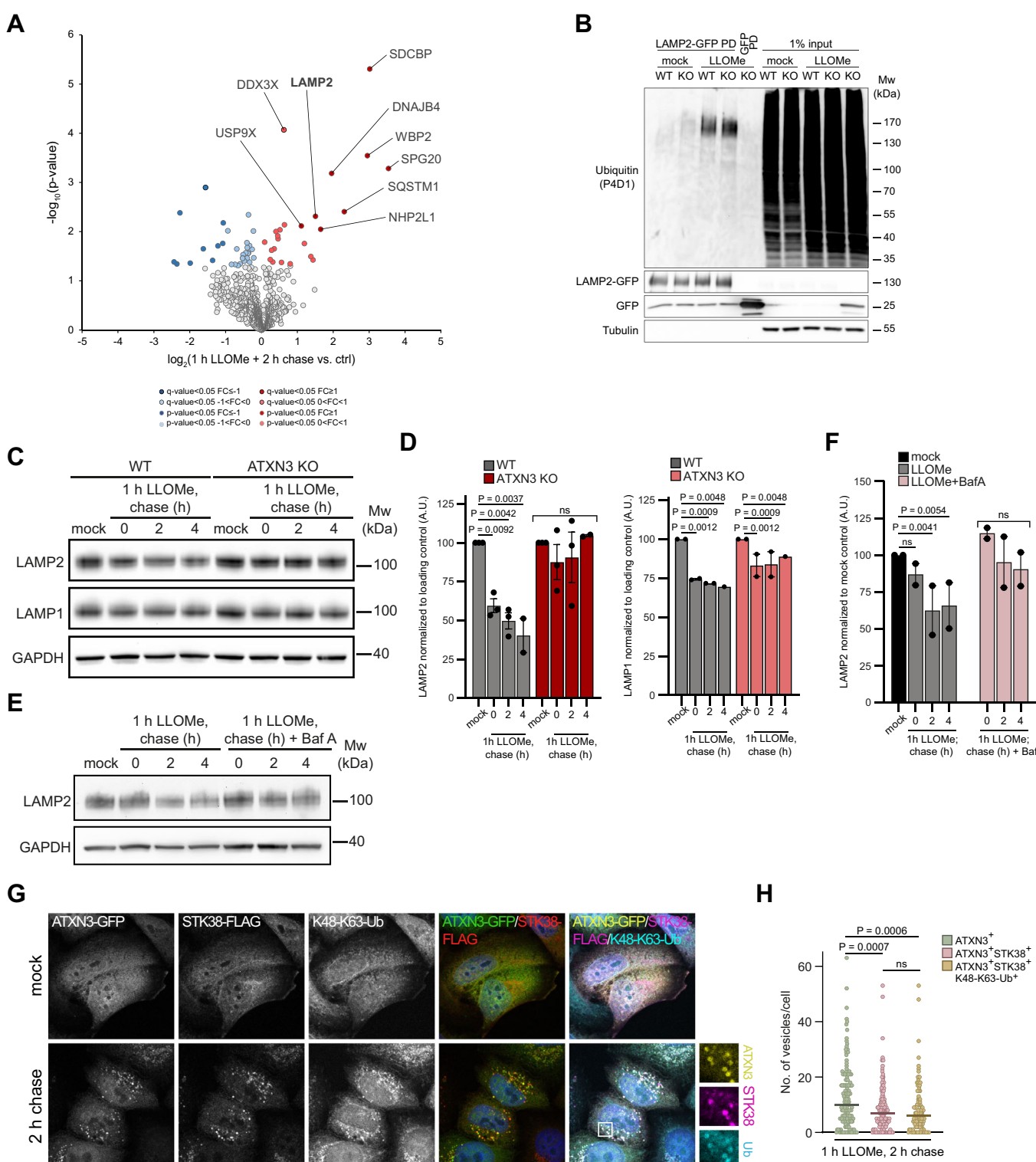

function does not appear to be directly relevant in our setup that monitors acute stress of lysosomal damage, because we find that the formation of the phagophore in lysophagy is not severely affected in ATXN3-compromised cells. Thus, ATXN3 has an additional critical function in ensuring cellular homeostasis by restoring the integrity of the lysosomal system in stress conditions.

Our work links ATXN3 to the regeneration of lysosomes after damage that also affects lysophagy. Previous work revealed membrane sealing mechanisms by the ESCRT machinery and lipid transfer that decrease membrane permeability (Bohannon and Hanson, 2020; Hoyer et al, 2022; Yang and Tan, 2023; Zhen et al, 2021). However, it has become clear that lysosomes undergo

**Figure 6.  LAMP2 is modified with K48-K63-branched ubiquitin chains and degraded in lysosomes dependent on ATXN3.**

(A) K48-K63-branched ubiquitin conjugates were affinity-isolated with NbSL3.3Q from mock or LLOMe-treated cells. Proteins were identified by quantitative mass spectrometry. Statistical analysis was performed with a two-sided two-sample $t$-test with permutation-based FDR ($q$ value <0.05) as well as uncorrected $p$ value <0.05, $n = 4$. Significantly enriched proteins are labeled. (B) Parental or ATXN3 KO cells expressing LAMP2-GFP were mock or LLOMe-treated. LAMP2-GFP was isolated from lysates and analyzed with a ubiquitin antibody. (C) ATXN3 facilitates degradation of LAMP2 following lysosomal damage. HeLa WT or ATXN3 KO cells were mock or LLOMe-treated for 1 h. Samples were taken at the indicated time points and analyzed by western blot with the indicated antibodies. (D) Quantification of (C). $n = 3$ (except for 4 h chase sample and LAMP1 quantification $n = 2$) biologically independent experiments. Mixed effects analysis with uncorrected Fisher's LSD test. Error bars represent the mean with SEM. (E) Lysosomal damage-induced LAMP2 degradation following Bafilomycin A1 treatment. HeLa WT or ATXN3 KO cells were mock or LLOMe, or LLOMe and Bafilomycin A1-treated for 1 h. Samples were taken at the indicated time points and analyzed by western blot with the indicated antibodies. Bafilomycin A1 treatment was continued during chase. (F) Quantification of (E). $n = 2$ biologically independent experiments. Two-way ANOVA with Tukey's multiple comparison test. Error bars represent the mean with SEM. (G) ATXN3 and K48-K63-branched ubiquitin chains colocalize with microautophagy regulator STK38 on damaged lysosomes. Cells were mock or LLOMe-treated and stained with indicated antibodies. Scale bar, 15 μm. (H) Quantification of (G). $n = 3$ biological replicates with >25 cells per condition per experiment. One-way ANOVA with Tukey's multiple comparison test. The graph shows mean ± SD. Source data are available online for this figure.

additional processes of regeneration for full recovery of lysosomal function. On the one hand, this includes tubulation processes that either resemble autophagic lysosome regeneration (ALR) in regenerating damaged lysosomes (Yu et al, 2010) (Bhattacharya et al, 2023) or are driven by the Parkinson's disease-associated kinase LRRK2 (Bonet-Ponce et al, 2020). Conversely, for full recovery, regenerating lysosomes require invagination processes, termed microautophagy (Lee et al, 2020; Ogura et al, 2023). Microautophagy is critical to restore membrane homeostasis and to degrade individual proteins that might be denatured during membrane damage. We show that ATXN3 acts in regenerating lysosomes that are marked by PI(4,5)P2. While the exact role of ATXN3 and branched ubiquitin chains on regenerating lysosomes is not fully understood, we find that ATXN3 facilitates lysosomal degradation of a fraction of LAMP2 that likely helps restore membrane homeostasis during the damage response (Yoshida et al, 2017). Our data further support the notion that this degradation of LAMP2 and possibly other proteins, is mediated by micro-lysophagy, given the association with a major regulator of the process, STK38 (Ogura et al, 2023).

Mechanistically, we demonstrate that ATXN3 fulfils its function by targeting and processing ubiquitin conjugates with K48-K63-branched ubiquitin chains. K48-K63-branched chains have been established as a stress response signal only recently (Lange et al, 2024). Our work shown here now confirms the relevance of K48-K63-branched ubiquitin chains in stress signaling and puts these chains at the center of the lysosomal damage response. We find that ATXN3 cooperates with its partner, the multifunctional ubiquitin-directed AAA-ATPase VCP/p97, in processing K48-K63-branched ubiquitin conjugates. The preferred substrates for p97 are K48-linked ubiquitin conjugates (Bodnar and Rapoport, 2017; Olszewski et al, 2019). We showed earlier that, during the lysosomal damage response, p97 targets K48-linked ubiquitin conjugates to promote phagophore formation and lysophagy in cooperation with an alternative DUB, YOD1 (Kravic et al, 2022; Papadopoulos et al, 2017). In contrast, we find here that YOD1, unlike ATXN3, does not target K48-K63-branched chains. This finding aligns with the notion that different cofactor proteins engage p97 in distinct reactions (Buchberger et al, 2015), and that K48-K63-mediated regulation of lysosome regeneration is specific to ATXN3. It is therefore tempting to speculate that branched ubiquitin chains reprogram p97 to cooperate with ATXN3 on damaged lysosomes and facilitate degradation of proteins such as LAMP2 in the lysosome via microautophagy.

Importantly, our data highlight that regenerating lysosomes represent a considerable fraction of lysosomes during the damage response and that the underlying processes are regulated in unanticipated ways involving K48-K63-branched ubiquitin chains and their processing by ATXN3 and p97. With ATXN3 and p97 being linked with neurodegeneration (McLoughlin et al, 2020; Meyer and Weihl, 2014), our findings underscore the relevance of this part of the lysosomal damage response for maintaining cellular homeostasis.

## Methods

### Reagents and tools table

| Reagent/resource | Reference or source | Identifier or catalog number |
|---|---|---|
| **Experimental models** | | |
| Human: HeLa Kyoto | Prof. Shuh Narumiya (Kyoto University) | RRID:CVCL_1922 |
| Human: U2OS | N/A | RRID:CVCL_0042 |
| Human: ARPE19 | ATCC | CRL-2302; RRID:CVCL_0145 |
| Human: HeLa ATXN3 KO | This paper | N/A |
| Human: U2OS ATXN3$^{dTAG}$ | This paper | N/A |
| Human: HeLa LAMP1-BFP | This paper | N/A |
| HeLa TMEM192-mKeima | Shima et al, 2023 | N/A |
| Bacteria: *Escherichia coli* BL21 (DE3) | New England Biolabs | C2527I |
| Bacteria: *Escherichia coli* XL1-Blue Competent Cells | Agilent | 200249 |
| **Recombinant DNA** | | |
| pcDNA5FRT/TO-Ataxin-3-Strep-HA | Hulsmann et al, 2018 | Addgene #113496 |
| pEGFP-N1-ATXN3 | This Paper | N/A |
| pmCherry-N1-ATXN3 | This Paper | N/A |
| pEGFP-N1-ATXN3-C14A | This Paper | N/A |
| pBlueScript-SK(+)-ATXN3 Homology Arms | This Paper | N/A |
| pCRIS-PITChv2-Puro-dTAG-BRD4 | Nabet et al, 2018 https://doi.org/10.1038/s41589-018-0021-8 | Addgene #91793 |
| pBlueScript-SK(+)-ATXN3 Homology Arms-PuroR-P2A-HA-dTAG | This Paper | N/A |

| Reagent/resource | Reference or source | Identifier or catalog number |
|---|---|---|
| GFP-C1-PLCdelta-PH | Stauffer et al, 1998 https://doi.org/10.1016/s0960-9822(98)70135-6 | Addgene #21179 |
| pcDNA3-Puro-LAMP1-eBFP2 | This Paper | N/A |
| pEGFP-N1-LAMP2 | This Paper | N/A |
| pmCherry-C2-TECPR1 | This Paper | N/A |
| pECMV-flag-STK38 | Devroe et al, 2004 https://doi.org/10.1074/jbc.M401999200 | N/A |
| **Antibodies** | | |
| anti-ALIX | BioLegend | 634501 |
| anti-Ataxin-3 | BioLegend | 650402 |
| anti-FLAG | Sigma | F3165 |
| anti-Ataxin-3 CoraLite Plus 488-conjugated | Proteintech | CL488-67057 |
| anti-Galectin-3 | Santa Cruz Biotechnology | sc-23938 |
| anti-LAMP1 | Santa Cruz Biotechnology | sc-20011 |
| anti-LAMP1 | Cell Signaling | D2D11 |
| anti-LC3 | MBL | PM036 |
| anti-LAMP2 | Santa Cruz Biotechnology | sc-18822 |
| anti-CNN2 | Thermo Fisher Scientific | PA5-61878 |
| anti-p62/SQSTM1 | Abnova | H00008878-M01 |
| anti-GFP | Santa Cruz Biotechnology | sc-9996 |
| anti-GFP | Roche | 11814460001 |
| anti-IST1 | Proteintech | #51002-1-AP |
| anti-IST1 | Proteintech | #66989 |
| anti-Tubulin | Sigma | T-5168 |
| anti-P4D1 | Cell Signaling Technology | 3936; RRID:AB_331292 |
| anti-GAPDH | Sigma | G8795 |
| horseradish peroxidase (HRP)-conjugated goat anti-rabbit IgG (WB, 1:10000) | Bio-Rad | 170-6515; RRID:AB_11125142 |
| horseradish peroxidase (HRP)-conjugated goat anti-mouse IgG (WB, 1:10000) | Bio-Rad | 1706516; RRID:AB_11125547 |
| Alexa Fluor-conjugated goat anti-rabbit, Alexa Fluor™ 568 (IF, 1:500) | Invitrogen | A11011; RRID:AB_143157 |
| Alexa Fluor-conjugated goat anti-rabbit, Alexa Fluor™ 488 (IF, 1:500) | Life Technologies | A11034; RRID:AB_2576217 |
| Alexa Fluor-conjugated goat anti-rabbit, Alexa Fluor™ 633 (IF, 1:500) | Thermo Fisher Scientific | A21071; RRID:AB_2535732 |
| Alexa Fluor-conjugated goat anti-mouse, Alexa Fluor™ 488 (IF, 1:500) | Thermo Fisher Scientific | 10696113 |
| Alexa Fluor-conjugated goat anti-mouse, Alexa Fluor™ 594 (IF, 1:500) | Thermo Fisher Scientific | A-11032; RRID:AB_2534091 |
| Alexa Fluor-conjugated goat anti-mouse, Alexa Fluor™ 633 (IF, 1:500) | Thermo Fisher Scientific | 10246252 |
| Alexa Fluor-conjugated goat anti-rat, Alexa Fluor™ 488 (IF, 1:500) | Thermo Fisher Scientific | A-11006; RRID:AB_2534074 |
| Alexa Fluor-conjugated goat anti-rat, Alexa Fluor™ 568 (IF, 1:500) | Thermo Fisher Scientific | A-11077; RRID:AB_2534121 |
| Alexa Fluor-conjugated goat anti-rat, Alexa Fluor™ 633 (IF, 1:500) | Thermo Fisher Scientific | A-21094; RRID:AB_2535749 |

| Reagent/resource | Reference or source | Identifier or catalog number |
|---|---|---|
| NbSL3.3Q Nanobody | Lange et al, 2024 | n.a. |
| **Oligonucleotides and other sequence-based reagents** | | |
| sgRNA ATXN3^dTAG (ACTCACTTTCTCGTGGAAGA) | This study, Microsynth | - |
| Ataxin-3 Double Nickase Plasmid | Santa Cruz | sc-417498-NIC |
| ATXN3 gPCR1 (5'-CCCCGTCTCCCACACAATTTA-3') | This study, Microsynth | - |
| ATXN3 gPCR2 (5'-CAGCAGGCTAGGCAGACTAC-3') | This study, Microsynth | - |
| ATXN3 seq1 (5'-TTCACTCGCTCTTCGCTTCA-3') | This study, Microsynth | - |
| ATXN3 seq2 (5'-AGTTCTTGCAGCTCGGTGAC-3') | This study, Microsynth | - |
| ATXN3 seq3 (5'-ATCATCCCACCACATGCCAC-3') | This study, Microsynth | - |
| ATXN3 seq4 (5'-AAGCGATGGAAAGTGACGGA-3') | This study, Microsynth | - |
| siATXN3 #1 UGGCAGAAGGAGGAGUUACTT | Wang et al, 2012 | - |
| siATXN3 #2 CAGGGCUAUUCAGCUAAGUAUTT | Sacco et al, 2014 | - |
| siATXN3 #3 GCACUAAGUCGCCAAGAAATT | Ashkenazi et al, 2017 | - |
| siATXN3 #4 GCAGGGCUAUUCAGCUAAGUT | Ashkenazi et al, 2017 | - |
| **Chemicals, enzymes and other reagents** | | |
| DMEM | PAN-Biotech | P04-03590 |
| DMEM/F12 | PAN-Biotech | P04-41150 |
| FBS | PAN-Biotech | P30-3306 |
| DPBS | PAN-Biotech | P04-36500 |
| Penicillin/Streptomycin | PAN-Biotech | P06-07100 |
| Imaging medium | PAN-Biotech | P04-03591 |
| Trypsin-EDTA | PAN-Biotech | P10-23100 |
| Lipofectamine 2000 | Thermo Fisher Scientific | 11668019 |
| Lipofectamine RNAiMax | Thermo Fisher Scientific | 13778150 |
| dTAG^VHL | Tocris | 6914 |
| CB-5083 | Selleckchem | S8101 |
| AlPcS2a | Frontier Scientific | P40632 |
| Benzonase | Merck | 70746-4 |
| HindIII-HF | New England Biolabs | R3104S |
| BamHI-HF | New England Biolabs | R3136S |
| XhoI | New England Biolabs | R0146S |
| N-Ethylmaleimide (NEM) | Sigma-Aldrich | E3876-5G |
| Puromycin | Sigma | P8833-100MG |
| L-Leucyl-L-Leucine methyl ester hydrobromide (LLOMe) | Sigma | L7393 |
| Chloroquine diphosphate salt | Merck/Sigma-Aldrich | C6628-50G |
| Bafilomycin A1 | Biomol | 110038 |
| Lysotracker Deep Red | Thermo Fisher Scientific | L12492 |
| TRIzol Reagent | Thermo Fisher Scientific | 15596026 |

| Reagent/resource | Reference or source | Identifier or catalog number |
|---|---|---|
| Complete EDTA-free Protease Inhibitor Cocktail | Roche | 04693132001 |
| SuperSignal West Pico Chemiluminescent substrate | Pierce | 15669364 |
| ECL Prime Western Blotting Detection Reagent | Amersham | 12994780(RPN2236) |
| IPTG (Isopropyl-β-D-thiogalactopyranosid) | VWR | A1008.0100 |
| AF 568 NHS-Ester | Lumiprobe | 24820 |
| ProLong Gold | Thermo Fisher Scientific | P36930 |
| BC Assay Protein Quantification Kit | VWR | 733-1404 |
| SuperScript II Reverse Transcriptase | Thermo Fisher Scientific | 18064014 |
| Oligo (dT) 12-18 Primer 0,5 μg/μl | Thermo Fisher Scientific | 18418012 |
| Gibson Assembly Kit | Thermo Fisher Scientific | A46627 |
| DQ-BSA | Invitrogen | D12051 |
| Bouin's solution | Carl Roth | 6482.3 |
| NHS-activated Sepharose 4 Fast Flow | VWR | 17-0906-01 |
| **Software** | | |
| Fiji | NIH | http://fiji.sc RRID:SCR_002285 |
| Adobe Photoshop | Adobe Inc. (2021) | https://www.adobe.com/products/photoshop.html RRID:SCR_014199 |
| Adobe Illustrator | Adobe Inc. (2021) | http://www.adobe.com/products/illustrator.html RRID:SCR_010279 |
| CellProfiler software 44 (4.2.5) | Broad Institute; Stirling et al, 2021 | https://cellprofiler.org/ RRID:SCR_007358 |
| CellProfiler software 44 (2.1.1) | Broad Institute | https://cellprofiler.org/ RRID:SCR_007358 |
| cyto2 model | n.a. | Cutler et al, 2022 |
| Kaluza (version 2) | Beckman Coulter | RRID: SCR_016182 |
| Leica Application Suite X | Leica Microsystems | https://www.leica-microsystems.com/products/microscope-software/details/product/leica-las-x-ls/ RRID:SCR_013673 |
| Andor iQ | Oxford Instruments | http://www.andor.com/scientific-software/iq-live-cell-imaging-software RRID:SCR_014461 |
| Excel (2021) | Microsoft Corporation | https://office.microsoft.com/excel RRID: SCR_016137 |
| SoftMax Pro | Molecular Devices | RRID:SCR_014240 |
| MaxQuant (1.6.0.1) | Cox and Mann, 2008; Cox et al, 2011 | RRID:SCR_014485 |
| GraphPad Prism 9.0 | GraphPad Software (San Diego, USA) | www.graphpad.com RRID:SCR_002798 |
| **Other** | | |
| - | - | - |

## Plasmids

pEGFP-N1-ATXN3 and pmCherry-N1-ATXN3 plasmids were created from the plasmid pcDNA5FRT/TO-Ataxin-3-Strep-HA (Hulsmann et al, 2018) using Gibson cloning. To generate the catalytically inactive form of ATXN3, site-directed mutagenesis was performed to obtain pEGFP-N1-ATXN3-C14A. To generate the p97/VCP binding mutant of ATXN3, site-directed mutagenesis was performed to obtain pEGFP-N1-ATXN3-VBM*. Homology arms for the N-terminus of ATXN3 (with silent mutations to prevent sgRNA binding) were ordered as a gBlock (Integrated DNA Technologies) containing HindIII and BamHI restriction sites. The gBlock was cloned into the pBlueScript-SK(+) vector by restriction digestion. PuroR-P2A-HA-dTAG from pCRIS-PITChv2-Puro-dTAG-BRD4 plasmid (gift from James Bradner; Addgene #91793) was inserted into the homology arms by Gibson cloning. Nanobody expression plasmids were described before (Lange et al, 2024). GFP-C1-PLCdelta-PH plasmid was a gift from Tobias Meyer (Addgene #21179). pEGFP-C1-p97 E578Q plasmid was described before (Kravic et al, 2022). LAMP1 was cloned into the pcDNA3-eBFP2 vector by amplifying human LAMP1 from HeLa cDNA using oligos containing HindIII and BamHI sites. pEGFP-N1-LAMP2 was cloned by the amplification of LAMP2 ORF from SH-SY5Y cDNA using restriction enzymes KpnI and BamHI. pmCherry-C2-TECPR1 was cloned by the amplification of TECPR1 ORF from SH-SY5Y cDNA using restriction enzymes XhoI and HindIII. pECMV-flag-STK38 was ordered from Addgene (#8927). The ORF of ULK1 was amplified by PCR and subcloned into pEGFP-C3 using HindIII and BamHI restriction enzymes.

## Cell lines

HeLa ATXN3 knockout cells were generated by transfection with Ataxin-3 Double Nickase Plasmid (Santa Cruz; sc-417498-NIC) according to the manufacturer's instructions. U2OS ATXN3dTAG cells were generated using sgRNA (ACTCACTTTCTCGTGGAAGA) targeting the N-terminus of ATXN3 using CRISPR Cas9 technology to allow integration of PuroR-P2A-HA-dTAG (derived from pCRIS-PITChv2-Puro-dTAG-BRD4, Addgene #91793). Cells were cultured in DMEM containing 10% fetal bovine serum (FBS, PAN-Biotech), 1% Penicillin/Streptomycin (PAN-Biotech), and selection antibiotic Puromycin (1 μg/ml). Integration was confirmed by Sanger sequencing following genomic PCR. To amplify the N-terminus of ATXN3 from genomic DNA in order to confirm the integration, the following primers were used: 5'-CCCCGTCTCCCACACAATTTA-3' and 5'-CAGCAGGCTAGGCA-GACTAC-3'. The primers 5'-TTCACTCGCTCTTCGCTTCA-3', 5'-AGTTCTTGCAGCTCGGTGAC-3', 5'-ATCATCCCACCACATGCC-AC-3', 5'-AAGCGATGGAAAGTGACGGA-3' were used for Sanger sequencing, and integration of PuroR-P2A-HA-dTAG at the N-terminus of ATXN3 was confirmed. The knock-in was also confirmed via Western blot. For the generation of HeLa LAMP1-BFP stable cell line, HeLa Kyoto cells were transfected with circular pcDNA3-Puro-LAMP1-eBFP2 and selected using Puromycin (1 μg/ml). Cell lines were not authenticated.

## Cell culture

HeLa and U2OS cells were cultured in DMEM containing 10% FBS and 1% Penicillin/Streptomycin. ARPE19 and SH-SY5Y cells were

cultured in DMEM:F12 supplemented with 10% FBS and 1% Penicillin/Streptomycin. Cells were grown in standard conditions, 37 °C and 5% $CO_2$. Cells were transfected with plasmids using Lipofectamine 2000 (Thermo Fisher Scientific) or with 10 nM siRNA using RNAiMAx (Thermo Fisher Scientific) according to the manufacturer's instructions. Transfected cells were analyzed after 24 h (plasmids) or 48 h (siRNA).

## RNA interference

siRNAs targeting ATXN3 were purchased from Microsynth:

siATXN3 #1 UGGCAGAAGGAGGAGUUACTT (Wang et al, 2012), siATXN3 #2 CAGGGCUAUUCAGCUAAGUAUTT (Sacco et al, 2014), siATXN3 #3 GCACUAAGUCGCCAAGAAATT (Ashkenazi et al, 2017), and siATXN3 #4 GCAGGGCUAUUCAG-CUAAGTT (Ashkenazi et al, 2017). siSTK38 CGUCGGC-CAUAAACAGCUTT (Ogura et al, 2023).

## Cell treatments

To induce lysosomal damage, cells were treated with 1 mM L-leucyl-L-leucine methyl ester hydrobromide (LLOMe, Sigma, # L7393) or, in the case of HeLa TMEM192-mKeima, with 500 μM LLOMe for the indicated times. For p97 inhibition, cells were treated with 5 μM CB-5083 (Selleckchem, #S8101) for the indicated times. Cells were treated with 100 nM Bafilomycin A1 (Biomol, #110038) or 100 μM Chloroquine diphosphate salt (Merck/Sigma-Aldrich, # C6628-50G) for the indicated times. To induce degradation of ATXN3$^{dTAG}$, cells were treated with 1 μM dTAG$^{VHL}$ (Torcis, #6914) for the indicated times. To inhibit canonical autophagy, cells were treated with 1 μM MRT68921 and 10 μM SAR405 for the indicated times.

## Denaturing immunoprecipitation

Cells were lysed with 100 μl of SDS-containing buffer (50 mM Tris pH 7.5, 1% SDS, 2.5 mM NEM) for 10 min on ice, scraped and SDS was diluted with 900 μl dilution buffer (50 mM Tris pH 7.5, 150 mM NaCl, 0.5 mM EDTA, 0.5% NP40 and freshly added 5 mM DTT, protease and phosphatase inhibitors). In order to remove DNA, 0.5 μl Benzonase was added and lysates were left on a rotor in a cold room for 50 min. Lysates were centrifuged at $17,000 \times g$ for 5 min. 1 ml of lysate was mixed with 50 μl of K48-K63 branched Ubiquitin nanobody beads slurry for the branched ubiquitin pull down (1.5 h in a cold room). Beads were washed three times with dilution buffer without SDS, boiled in 15 μl 2x Laemmli buffer, and used for SDS–PAGE gel electrophoresis.

## K48-K63 branched pulldown and processing for MS

Denaturing immunoprecipitation was performed as described above. The eluate was further processed for MS using the single-pot, solid-phase-enhanced sample-preparation (SP3) technology. In brief, para-magnetic Sera-Mag SpeedBeads A and B (Cytiva) were added in a 1:1 ratio to the eluate, and acetonitrile was added to a final concentration of 70%. After incubation at 24 °C for 30 min at 1000 rpm, the supernatant was removed and the beads reconstituted in 50 mM ABC. After subsequent reduction with 30 mM DTT (30 min, 24 °C, 1200 rpm), alkylation with 400 mM chloroacetamide (30 min, 42 °C, 1200 rpm), and quenching with 30 mM DTT, acetonitrile was added to a final

concentration of 70% for 30 min at 24 °C, shaking at 1200 rpm. Beads were then washed twice with 100% acetonitrile, twice with 80% ethanol, and dried. Beads were resuspended in digestion mix (0.03 μg/μL Trypsin, 0.015 μg/μL Lys-C, 50 mM ABC) and incubated overnight at 37 °C. The next day, digest was stopped by the addition of 8% FA, and peptides were separated from the beads with Spin-X columns (Corning CoStar). After drying, peptides were resolved in 0.1% formic acid.

## MS data collection and analysis

Samples were separated using an Easy-nLC1200 liquid chromatograph (Thermo Scientific), followed by peptide detection on a Q Exactive HF mass spectrometer (Thermo Scientific). Peptides were loaded onto 75 μm × 15 cm custom-made fused silica capillaries packed with C18AQ resin (Reprosil-PUR 120, 1.9 m, Dr. Maisch HPLC). Peptide mixtures were separated on a 35 min buffer B (80% ACN, 0.1% FA) gradient (10–38% for 23 min, 38–60% for 2 min, 60–100% for 2 min, 100% for 3 min, 100%–5% for 2 min, and 5% for 3 min) at a flow rate of 400 nl/min. Peptides were ionized using a Nanospray Flex Ion Source (Thermo Scientific) and identified in full MS/ddMS² mode. The top 15 most intense peaks from each full MS scan were selected for subsequent fragmentation, with a dynamic exclusion of 20.0 s, excluding peptides with unassigned charge or charges of 1 or >8. MS1 resolution was set to 60,000 with a scan range of 300–1650 m/z, MS2 to 15,000. AGC target1 was set to 3e6, AGC target2 to 1e5.

Data collection was controlled by Tune/Xcalibur (Thermo Scientific). Raw data files from quadruplicate samples were analyzed using MaxQuant (1.6.0.1) Andromeda search engine in reversed decoy mode based on a human reference proteome (Uniprot-FASTA, UP000005640, downloaded November 2024) with an FDR of 0.01 on the protein level. Digestion parameters were set to specific digestion with trypsin/p with a maximum number of two missed cleavage sites and a minimum peptide length of 7. Methionine oxidation (15.994946) and N-terminal protein acetylation (42.010565) were set as variable modifications, and carbamidomethylation of cysteine as a fixed modification. The peptide mass tolerance was set to 20 ppm (first search) and to 4.5 ppm (main search). Label-free quantification (with a minimum ratio count set to 2), re-quantification, and match-between runs was selected.

Resulting text files from MaxQuant analysis were further processed in Perseus (version 1.6.14.0). Briefly, common contaminants, site-specific, and reverse identifications were excluded. Protein groups with less than two peptides and MS/MS counts were removed. Furthermore, data were filtered to keep only protein groups present in at least three out of four replicates per condition. LFQ intensities were $\log_2$ transformed, and remaining missing values were imputed with random numbers from a normal distribution with 0.3 width and 1.8 down shift calculated separately for each sample. Statistical analysis was performed with a two-sided two-sample $t$-test with permutation-based FDR ($q$ value <0.05) as well as uncorrected $p$ value <0.05. Protein groups' sites were considered regulated when reaching a $\log_2$ fold change $\geq 1$ or $\leq -1$ and passing the significance threshold $q < 0.05$.

## Antibodies and nanobody

The following primary antibodies were used in the study: anti-ALIX (mouse, BioLegend, 634501; IF 1:250), anti-Ataxin-3 (mouse, BioLegend, 650402; WB 1:1000), anti-FLAG (mouse, Sigma,

F3165, IF 1:110), anti-Ataxin-3 CoraLite Plus 488-conjugated (mouse, Proteintech, CL488-67057; IF 1:500), anti-Galectin-3 (rat, Santa Cruz Biotechnology, sc-23938; IF 1:500), anti-LAMP1 (mouse, Santa Cruz Biotechnology, sc-20011; IF 1:500), anti-LAMP1 (rabbit, Cell Signaling, D2D11; IF 1:500), anti-LC3 (rabbit, MBL, PM036; IF 1:500), anti-LAMP2 (mouse, Santa Cruz Biotechnology, sc-18822; WB 1:1000), anti-CNN2 (rabbit, Thermo Fisher Scientific, PA5-61878; WB 1:1000), anti-p62/SQSTM1 (mouse, Abnova, H00008878-M01; IF 1:500), anti-GFP (mouse, Santa Cruz Biotechnology, sc-9996; IF 1:500), anti-GFP (mouse, Roche, 11814460001; IF 1:500), anti-IST1 (rabbit, Proteintech, #51002-1-AP, IF 1:500), anti-IST1 (mouse, Proteintech, #66989, IF 1:500), anti-STK38 (mouse, Santa Cruz, sc-365555; WB 1:1000), anti-Tubulin (mouse, Sigma, T-5168; WB 1:8000), anti-GAPDH (mouse, Sigma, G8795; WB 1:2000). HRP-coupled secondary antibodies were purchased from Bio-Rad and Alexa Fluor-conjugated secondary antibodies from Invitrogen.

NbSL3.3Q nanobody was expressed in *E. coli* BL21 and purified as described (Lange et al, 2024). The purified nanobody was labeled with Alexa Fluor 568 NHS ester (Lumiprobe; eightfold molar excess) and repurified using a Superdex 75 10/300 column (GE Healthcare). For pulldowns, NbSL3.3Q protein was coupled to amine-reactive agarose beads, as previously described (Lange et al, 2024).

## Immunofluorescence staining and confocal laser-scanning microscopy

Cells were cultured on coverslips, fixed in 4% paraformaldehyde at room temperature or in 100% methanol at −20 °C (for NbSL3.3Q staining) or in 1:1 Bouin:PBS with 4% Sucrose for 30 min at room temperature (for endogenous Ataxin-3 staining with ProetinTech antibody), permeabilized with 0.1% Triton X-100 in PBS and blocked in PBS with 3% BSA + 0.1% Triton X-100 + 0.1% saponin. Following immunofluorescence staining, cells were mounted in ProLong Gold (Thermo Fisher Scientific).

Confocal laser-scanning microscopy was performed on a Leica TCS SP8X Falcon confocal microscope (Leica Microsystems), equipped with HyDs SMD detectors, HC PL APO 63x/1.4 Oil CS2 objective, white-light laser, Argon laser, and a 405 diode laser, and the Leica Application Suite X (LAS-X) software version 3.5.7. Images were acquired in a 1024 × 1024 ox format (1.5x zoom) with a bit depth of 12 bit.

In addition, imaging was performed on a TCS SP8 HCS A confocal microscope using the LAS-X software (3.5.7.23225). An HC PL APO 63×/1.4 NA CS2 oil immersion objective was used to acquire the images. Fluorophores were excited with a 405 nm diode laser, an argon laser line 488 nm, a 561 nm diode-pumped solid-state laser, and a 633 nm helium-neon laser. Signals were detected with standard PMT detectors and sensitive Hybrid detectors (HyD) in a 2048 × 2048 px format (1x zoom) with a bit depth of 12 bit.

## Live-cell microscopy, photodamage, and SIM

Live cell imaging was performed at 37 °C in imaging medium (P04-03591, PAN-Biotech) supplemented with 10% FCS and L-glutamine (11500626, Fisher Scientific). For the lysotracker fluorescence intensity experiment, cells were incubated with LysoTracker Deep Red (Thermo Fisher) for 1 h and washed twice with PBS before the addition of imaging medium. Prior to treatment with LLOMe, multiple regions were defined per sample for automated imaging using an Eclipse Ti-E (Nikon) inverted microscope equipped with an Andor AOTF Laser Combiner, a CSU-X1 Yokogawa spinning disk unit, and an iXon3 897 single photon detection EMCCD camera (Andor Technology). CSU 640 nm laser and CFI Apo TIRF 60x/1.49 oil immersion objective (Nikon) were used for image acquisition. Every region in each sample was imaged once before the addition of LLOMe and then every 5 min after LLOMe treatment to track the loss of lysotracker fluorescence intensity from lysosomes.

Spinning disk microscopy with laser-induced lysosomal damage was performed on the Eclipse Ti-E (Nikon) inverted microscope with an Andor ILE Laser Combiner, a CSU-X1 Yokogawa spinning disk unit, and a ZL41 Cell sCMOS camera (Andor Technology). Laser lines used for excitation of EGFP and mCherry were 488 and 561 nm, respectively. Images were acquired using a CFI APO TIRF 60x/1.49NA oil immersion objectives (Nikon). To induce LMP with light, cells were treated with 125 nM AlPcS2a (P40632, Frontier Scientific) as described before (Kravic et al, 2022). Damage was induced with a 640 nm laser using the Micropoint 4 Scanned Photo Stimulation device (Andor Technology). Image acquisition was controlled by Andor IQ3 Software (Andor Technology) as described before (Kravic et al, 2022).

HeLa TMEM192-mKeima cells were plated in an 8-well Ibidi μ-slide and transfected with the indicated siRNAs. After 48 h of knockdown, live cells were imaged before and after 1 h of treatment with 500 μM LLOMe using a Leica TCS SP8X Falcon confocal microscope (specifications above) at 37 °C in imaging medium. Cells were washed twice with PBS after LLOMe treatment, maintained in imaging medium for recovery, and imaged again after 6 h of chase.

Images were processed using Fiji software (https://imagej.net/Fiji), Adobe Photoshop, and Illustrator. Automated image analysis was done with CellProfiler software version 2.1 (Carpenter et al, 2006; Stirling et al, 2021). Excel 2016 (Microsoft Corporation) and GraphPad Prism 10.3 (GraphPad Software Inc.) were used for graphical representation and statistical analysis.

## Structured illumination microscopy (SIM)

Live HeLa LAMP1-BFP cells were imaged at 37 °C and 5% $CO_2$ in imaging medium (P04-03591, PAN-Biotech) supplemented with 10% FBS and 1% Penicillin/Streptomycin. Cells were transfected with a plasmid encoding ATXN3-GFP or GFP-PLC PH 1 day before imaging. The cells were loaded with LysoTracker Deep Red (LysoTr) for 10 min prior to imaging. Live cells were imaged once before LLOMe treatment, then treated with 1 mM LLOMe for 12 min or 1 h, washed twice with PBS, and incubated in imaging medium containing LysoTracker to track recovery of LysoTracker and recruitment of the overexpressed protein. Cells were imaged again after washout at the indicated time points.

For live cell imaging using 2D-SIM and 3D-SIM, images were acquired using a commercial Nikon N-SIM S microscope system equipped with an ORCA-Fusion BT sCMOS camera (Hamamatsu Photonics K.K.) using NIS-Elements software (Nikon). Images were acquired using a CFI SR HP Apochromat TIRF 100xAC (NA 1.49) oil immersion objective. ZIVA light engine (Lumencor) equipped with six solid-state laser lines was used as an excitation

light source. Laser lines used for excitation of Lysotracker Deep Red, GFP, and BFP were 637, 476, and 405 nm, respectively. Single-band bandpass emission filters FF01-460/60 for BFP, FF01-525/50 for GFP, and FF01-692/40 for Lysotracker Deep Red (Semrock) were used. Multipoint timelapse images were acquired in three individual z-planes per position with a z-spacing distance of 0.5 μm. 2D-SIM and 3D-SIM raw data were reconstructed using the Slice Reconstruction in NIS-Elements with default settings.

## Lysotracker washout analysis 2D-SIM

Automated image analysis of 2D-SIM LLOMe washout experiments was performed with CellProfiler software (version 4.2.6) (Stirling et al, 2021). Cells were identified using the Run Omnipose plugin for CellProfiler with the cyto2 model (Cutler et al, 2022). Accuracy of cell detection and, if necessary, correction of cell outlines was controlled by a manual editing step (EditObjectsManually). To enhance vesicular structures of ATXN3, LAMP1 and LysoTracker, feature enhancement using Difference of Gaussians (DoG) was applied to the respective channels. Vesicles were detected and related to the respective parent cells. Double-positive vesicles were detected using the MaskObjects module with a minimum overlap fraction of 0.2.

## Cell death analysis by flow cytometry

Cells were harvested by trypsinization, washed in PBS, incubated in 10 μg/ml propidium iodide (PI) for 15 min at room temperature, and 50,000 events/sample were acquired at the MACSQuant MQ16 flow cytometer. For cell death analysis, cells were gated for GFP expression, and the percentage of PI-positive cells was determined with Kaluza Analysis software.

## DQ-BSA assay

Cells were treated with DQ-BSA reagent (DQ™ Red BSA (Invitrogen™) Cat. No D12051) for 7 h at a concentration of 10 μg/ml. Prior to microscopy, cells were fixed in PFA, stained with DAPI, and mounted in ProLong Gold (Thermo Fischer Scientific). Samples were imaged using a Leica TCS SP8X Falcon confocal microscope (Leica Microsystems) as described above.

## Immunoblotting

Cells were lysed in lysis buffer consisting of 150 mM KCl, 50 mM Tris-HCl pH 7.4, 5 mM $MgCl_2$, 5% Glycerol, 1% Triton X-100, 2 mM ß-mercaptoethanol, and supplemented with protease inhibitors (complete EDTA-free protease inhibitor cocktail, Roche). Proteins were separated by SDS–PAGE and transferred to nitrocellulose membranes (Amersham, GE Healthcare). Immunoblot analysis was performed with the indicated antibodies and visualized with SuperSignal West Pico Chemiluminescent substrate (Pierce).

## Statistical analysis

All quantitative data were presented as the mean ± SD or SEM of biologically independent samples, or all data points from independent experiments are presented with the line indicating median, unless stated otherwise. The number of replicas are indicated in the figure legends and chosen according to the experimental setup. Statistical analysis was carried out using GraphPad Prism 10.3 software on the mean of independent experiments. One- or two-way ANOVA was used to determine statistical significance, unless stated otherwise. A $p$ value <0.05 was considered statistically significant. Image data was quantified automatically to avoid human bias. Blinding was not applied.

## Data availability

Source data for gels, western blots, and mass spectrometry, as well as raw quantification data, were included in this submission. Microscopy and flow cytometry source data is available online via the following links:

https://doi.org/10.17632/963gy85cx3.1
https://doi.org/10.17632/yk5fs9v2rj.1
https://doi.org/10.17632/b6r4gtrj3y.1
https://doi.org/10.17632/m62b7d3n8m.1
Reagents are available upon request.

The source data of this paper are collected in the following database record: biostudies:S-SCDT-10_1038-S44318-025-00517-x.

## Peer review information

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

## Acknowledgements

We acknowledge the use of equipment and the support by the Imaging Center Campus Essen (ICCE), and specifically thank J. Koch for his help. This research was funded by a joint grant of the Deutsche Forschungsgesellschaft (DFG, German research foundation) to HM and CB (Project-ID 447112704). HM, MK, and NS were supported by the DFG CRC 1430 (ID 424228829), and CB by EXC 2145 SyNergy (ID 390857198) and the CRC 1177 (ID 259130777). Microscopes were funded by DFG INST 20876/480-1 FUGG and INST 20876/294-1 FUGG. YK was funded by the ERC Consolidator grant (StressHUb grant 101002428) and MRC grant MC_UU_00038/3.

## Author contributions

**Maike Reinders**: Conceptualization; Data curation; Formal analysis; Investigation; Visualization; Methodology. **Bojana Kravic**: Conceptualization; Data curation; Formal analysis; Investigation; Visualization; Methodology. **Pinki Gahlot**: Conceptualization; Data curation; Formal analysis; Investigation; Visualization; Methodology. **Sandra Koska**: Formal analysis; Investigation. **Johannes van den Boom**: Investigation. **Nina Schulze**: Data curation; Investigation; Methodology. **Sophie Levantovsky**: Data curation; Formal analysis; Investigation. **Stefan Kleine**: Resources. **Markus Kaiser**: Conceptualization; Resources; Funding acquisition. **Yogesh Kulathu**: Conceptualization; Resources; Funding acquisition. **Christian Behrends**: Conceptualization; Resources; Data curation; Formal analysis; Supervision; Funding acquisition. **Hemmo Meyer**: Conceptualization; Data curation; Formal analysis; Supervision; Funding acquisition; Visualization; Writing—original draft; Project administration; Writing—review and editing.

Source data underlying the figure panels in this paper may have individual authorship assigned. Where available, figure panel/source data authorship is listed in the following database record: biostudies:S-SCDT-10_1038-S44318-025-00517-x.

## Funding

## Disclosure and competing interests statement

The authors declare no competing interests.

# Expanded View Figures

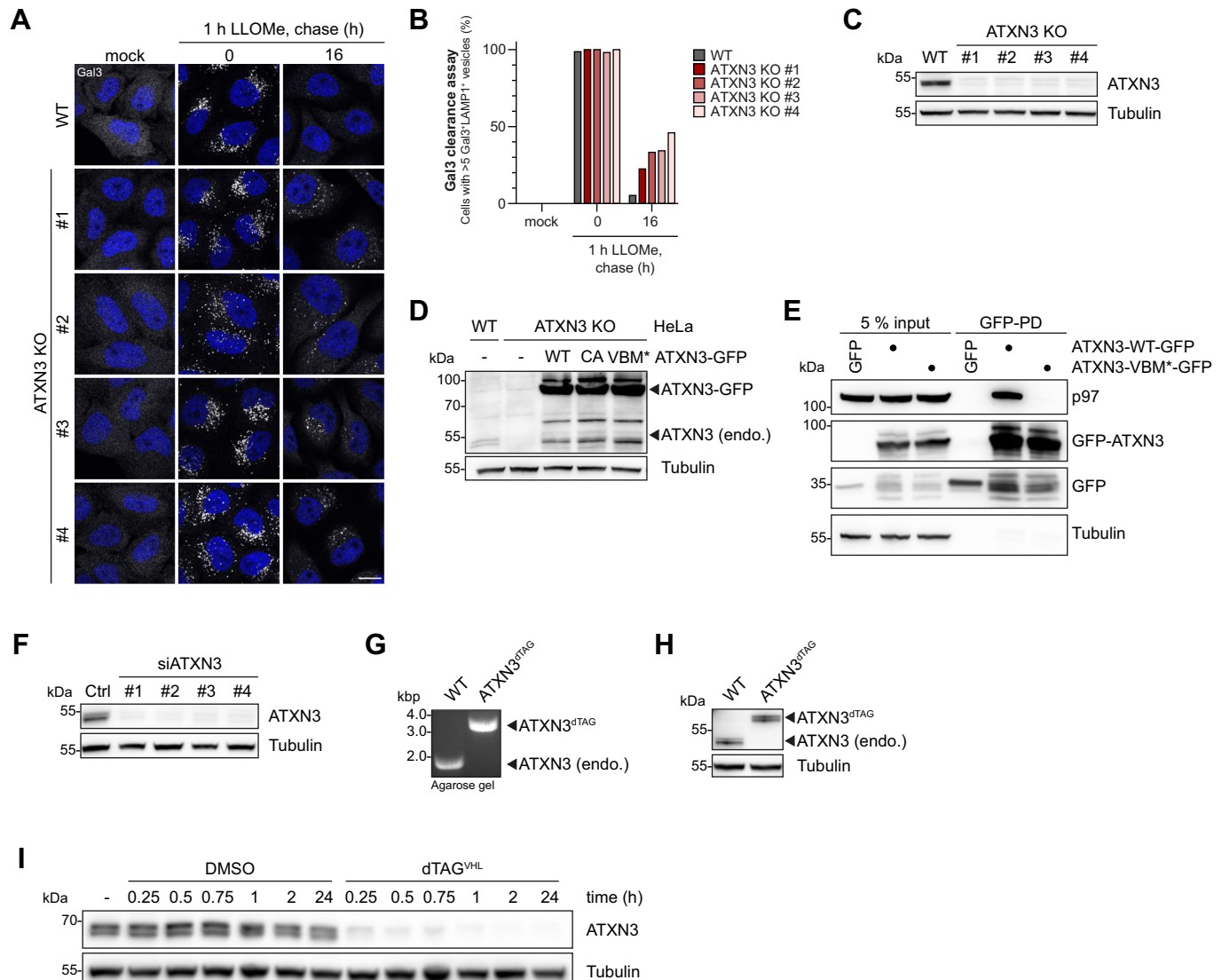

**Figure EV1.   (related to Fig. 2): ATXN3 is essential for the clearance of damaged lysosomes and the restoration of degradative compartments after lysosome damage.**

(**A**) Gal3 clearance assays in HeLa parental and 4 different ATXN3 KO clones performed as in Fig. 2A. Scale bar, 15 µm. (**B**) Quantification of (**A**), representative experiment with >40 cells per conditions. The graph shows the mean. (**C**) Western blot verification of ATXN3 KO. (**D**) Western blot verification of rescue constructs used in Fig. 2A. (**E**) ATXN3 mutant of the VBM motif (VBM*) is deficient in p97 binding. (**F**) Western blot evaluation of siRNA-mediated ATXN3 depletion. (**G**) PCR on genomic DNA confirming genomic insertion of the FKBP[F36V] tag in ATXN3[dTAG] U2OS cells. (**H**) Western blot analysis of lysates of U2OS WT and gene-edited ATXN3[dTAG] cells. (**I**) Western blot assessment of the induced degradation of ATXN3 [dTAG] in U2OS cells.

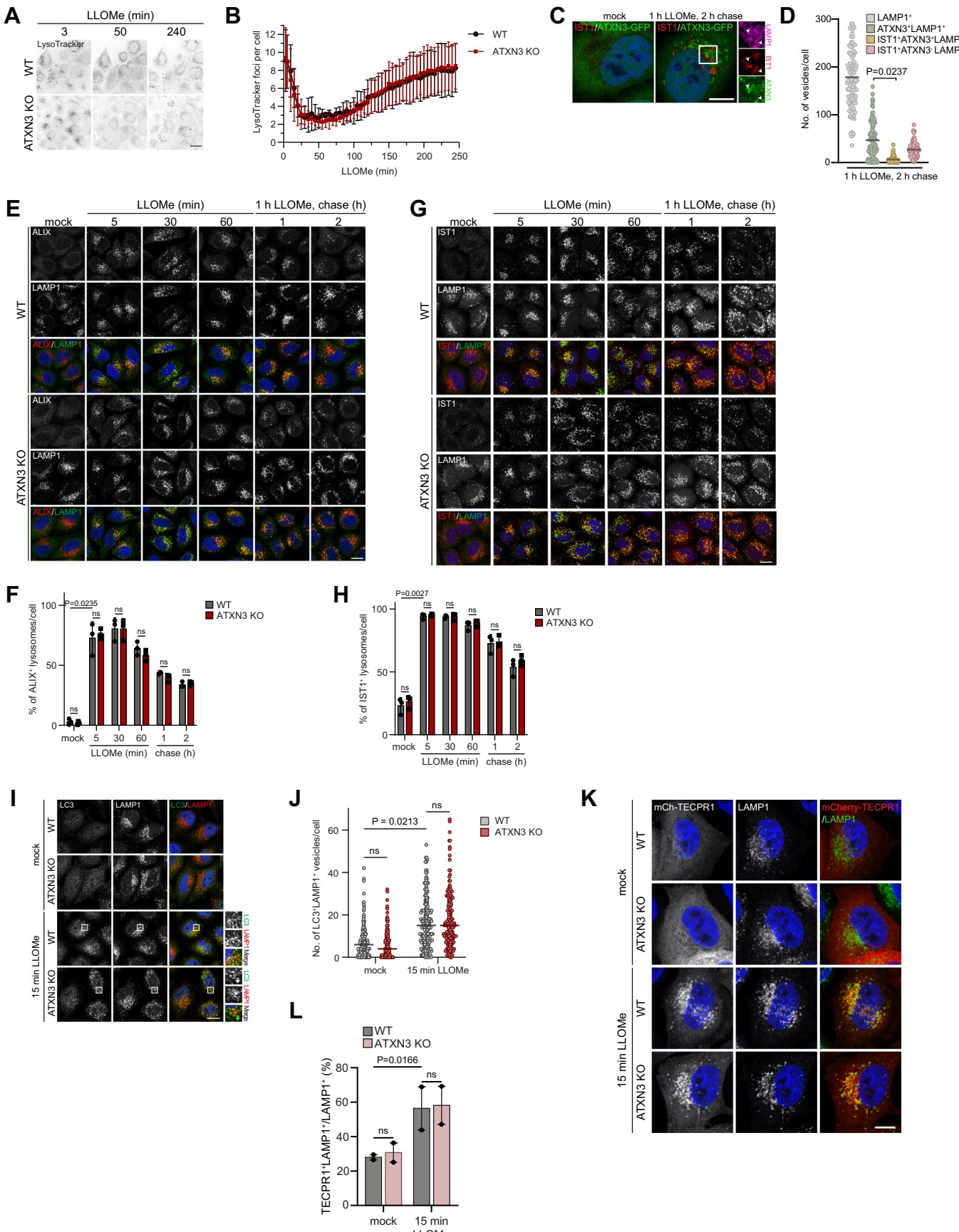

◀ **Figure EV2. ATXN3 is not involved in initial membrane repair pathways after lysosomal damage.**

(**A**) HeLa WT and ATXN3 KO cells were loaded with LysoTracker, treated with LLOMe, and LysoTracker recovery was imaged over the indicated period. Note that recovery of LysoTracker is unaffected in ATXN3 KO cells. Scale bar, 20 μm. (**B**) Quantification of (**A**), $n = 3$ biologically independent experiments with >70 cells quantified per condition per experiment. The graph shows mean ± SD. (**C**) Lack of colocalisation of ATXN3 and the ESCRT-III component IST1. HeLa cells expressing ATXN3-GFP were treated as indicated, fixed, and immuno-stained for IST1 and LAMP1. Arrowheads indicate ATXN3-positive and IST1-negative lysosomes. Scale bar, 10 μm. (**D**) Quantification of (**C**). $n = 3$ biological replicates with >30 cells per condition per experiment. One-way ANOVA with Tukey's multiple comparison test. The line indicates the mean. (**E**) ALIX localization is not affected in ATXN3 KO cells. HeLa WT or ATXN3 KO cells were mock- or LLOMe-treated and stained with the indicated antibodies. Scale bar, 15 μm. (**F**) Quantification of (**E**). $n = 3$ biological independent experiments with >40 cells per condition per experiment. Significance was tested by two-way ANOVA with Dunnett's multiple comparison test. The graph shows mean ± SD. (**G**) IST1 localization is not changed in HeLa ATXN3 KO compared to WT cells. Cells were treated as indicated, fixed, and immuno-stained for IST1 and LAMP1. Scale bar, 15 μm. (**H**) Quantification of (**G**). $n = 3$ biological replicates with >40 cells per condition per experiment. Two-way ANOVA with Dunnett's multiple comparison test. The graph shows mean ± SD. (**I**) Early LC3 recruitment indicates that CASM is not affected in ATXN3 KO cells. HeLa WT or ATXN3 KO cells were treated for 15 min with LLOMe and stained with the indicated antibodies. Scale bar, 15 μm. (**J**) Quantification of (**I**). $n = 3$ biologically independent experiments with >40 cells per condition per experiment. Significance was tested by two-way ANOVA with Fisher's LSD test. The graph shows mean ± SD. (**K**) Recruitment of TECPR1 is not compromised in ATXN3 KO cells. HeLa WT or ATXN3 KO cells transiently overexpressing mCherry-TECPR1 were mock- or LLOMe-treated for LLOMe for 15 min and stained for LAMP1. Scale bar, 10 μm. (**L**) Quantification of (**K**). $n = 2$ biologically independent experiments with 18 or more cells per condition per experiment. Two-way ANOVA with uncorrected Fisher's LSD test. The graph shows mean ± SEM.

 

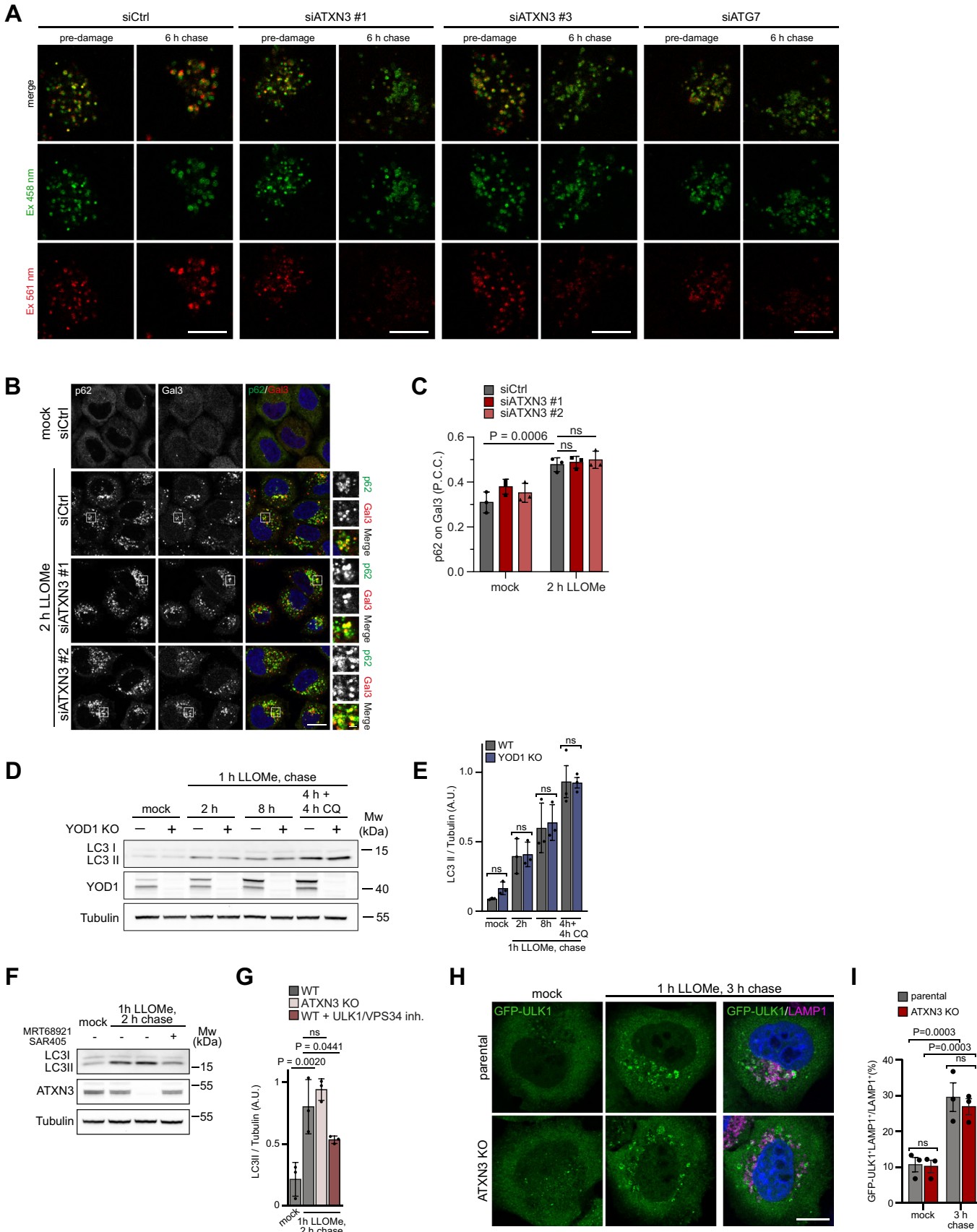

**Figure EV3.   (related to Fig. 3) ATXN3 is required for the completion of lysophagy.**

(A) Micrographs corresponding to the TMEM192-mKeima lysophagy assay in Fig. 3A as indicated. Scale bar, 10 μm. (B) ATXN3 depletion does not affect recruitment of SQSTM1/p62 to damaged lysosomes. HeLa cells treated with indicated siRNAs were mock or LLOMe-treated and fixed after 2 h. Cells were immuno-stained for Gal3 and p62. Note that these samples were also stained for LC3 (shown in Fig. 3B). Scale bar, 15 μm. (C) Quantification of (B), $n = 3$ biologically independent experiments with >70 cells quantified per condition per experiment. Two-way ANOVA with Tukey's multiple comparison test. The graph shows mean ± SD. (D) LC3 lipidation is not affected in YOD1 KO cells. (E) Quantification of (D). $n = 3$ biologically independent experiments. Two-way ANOVA with Tukey's multiple comparison test. Error bars represent the mean with SD. (F) Phagophore-associated LC3 lipidation during the lysosomal damage response is not affected by ATXN3 KO. Phagophore formation was inhibited by the indicated ULK1/2 and PI3KC3 inhibitors, and LC3-II formation was detected by Western blot analysis. Note the reduction of LC3-II after inhibition of phagophore formation, which is not observed in ATXN3 KO cells. (G) Quantification of (F). $n = 3$ biologically independent experiments. One-way ANOVA with Newman–Keuls multiple comparison test. Error bars represent the mean with SD. (H) ULK1 translocation to damaged lysosomes is not affected in ATXN3 KO cells. Micrographs show wildtype and ATXN3 KO cells transiently overexpressing GFP-ULK1 in control conditions and after 1 h LLOMe followed by 3 h chase. Scale bar, 10 μm. (I) Quantification of (H). $n = 3$ biologically independent experiments with >15 cells per experiment. Two-way ANOVA with Tukey's multiple comparison test. Error bars represent the mean with SEM.

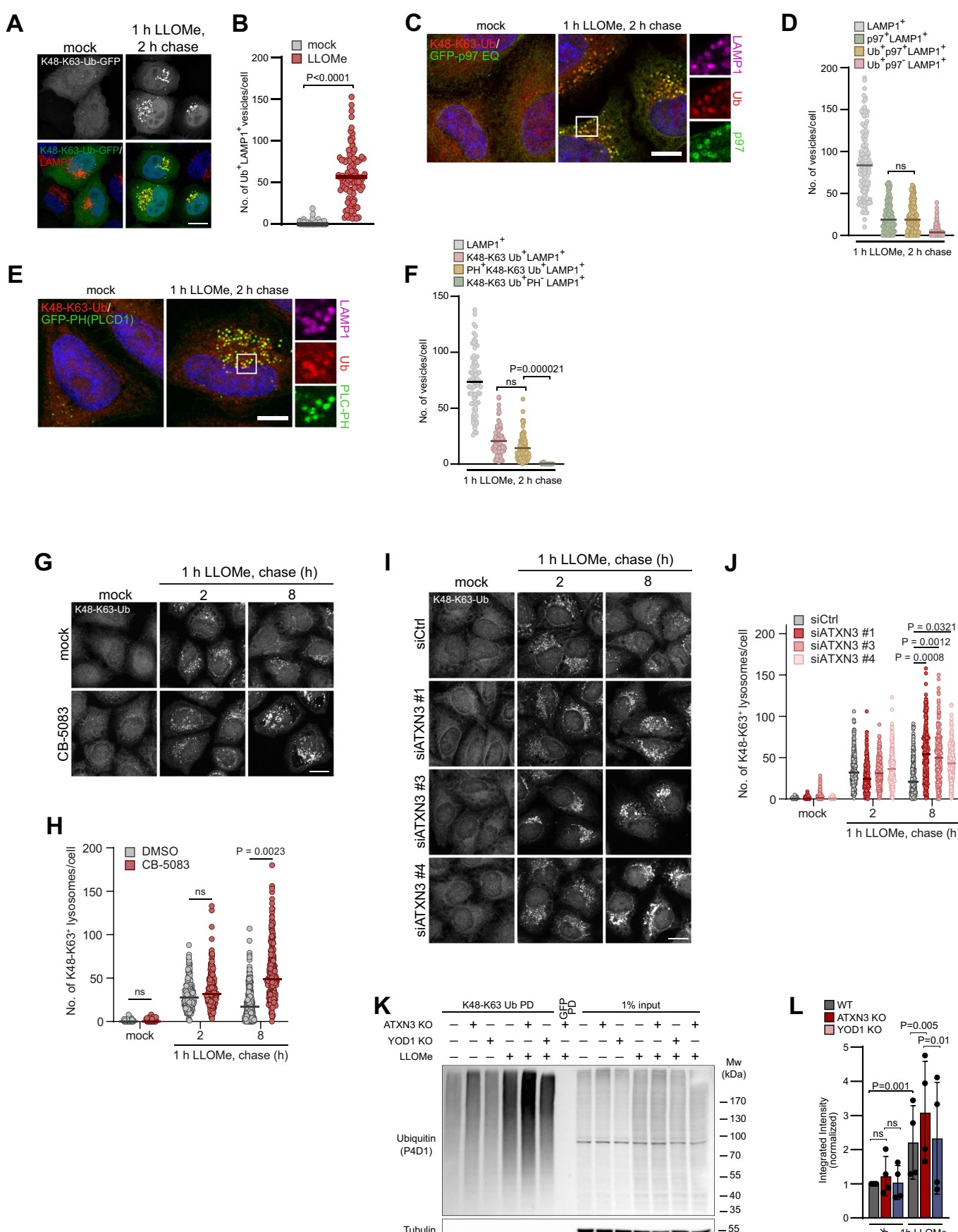

◄ **Figure EV4.** (related to Fig. 5). Damaged lysosomes are modified with K48-K63-branched ubiquitin chains.

(A) HeLa cells expressing NbSL3.3Q-GFP specific for K48-K63-branched chains were mock or LLOMe-treated, fixed, and immuno-stained for LAMP1. Note the prominent localization of NbSL3.3Q-GFP on lysosomes specifically after induction of lysosome damage. Scale bar, 15 μm. (B) Quantification of (A), $n = 3$ biological replicates with >20 cells per condition per experiment. Statistical significance was calculated with an unpaired $t$-test. Lines represent the mean. (C) HeLa cells expressing p97-E578Q were fixed and stained for K48-K63-branched ubiquitin chains with NbSL3.3Q. Scale bar, 10 μm. (D) Quantification of (C), $n = 3$ biological replicates with >30 cells per condition per experiment. One-way ANOVA with Tukey's multiple comparison test. The line indicates the mean. (E) K48-K63 branched chains accumulate on regenerating lysosomes. Scale bar, 10 μm. (F) Quantification of (E). $n = 2$ biologically independent experiments with >30 cells per condition per experiment. Lines represent the mean. Unpaired student's $t$-test. (G) HeLa cells were incubated with p97 inhibitor CB-5083 or vehicle alone and mock or LLOMe-treated. Cells were fixed at indicated time points and stained with NbSL3.3Q for K48-K63-branched ubiquitin chains. Scale bar, 15 μm. (H) Quantification of (E) $n = 3$ biological replicates with >70 cells per condition per experiment. Two-way ANOVA with Tukey's multiple comparison test. The line indicates the median. (I) Damage-induced K48-K63-branched chains persist in ATXN3-depleted cells. HeLa cells were treated with indicated siRNAs and K48-K63-branched chains persistence was monitored after lysosomal damage was assayed as in (G). Scale bar, 15 μm. (J) Quantification of (G). $n = 3$ biologically independent experiments with >30 cells quantified per condition per experiment. Two-way ANOVA with Tukey's multiple comparison test was used to test significance. The line indicates the median. (K) Damage-induced K48-K63-branched chains accumulate in ATXN3 KO cells, but not in YOD1 KO cells. (L) Quantification of (I). $n = 4$ biologically independent experiments. Two-way ANOVA with Uncorrected Fisher's LSD test. Error bars represent the mean with SD.

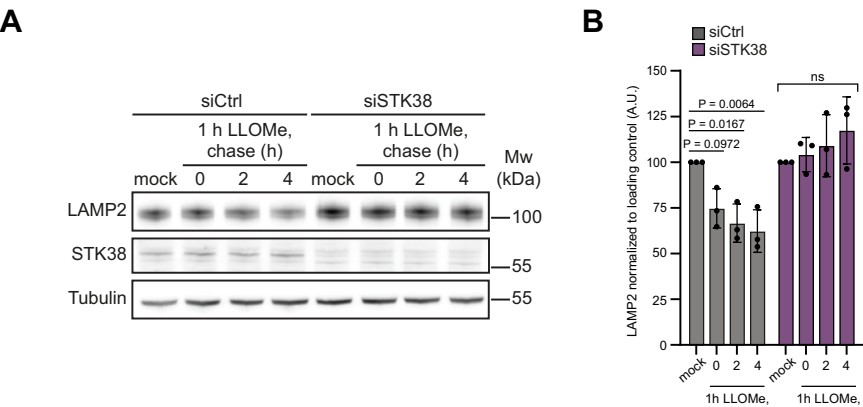

**Figure EV5.   (related to Fig. 6). LAMP2 is degraded through STK38-regulated microautophagy.**

(A) Knockdown of STK38 inhibits LAMP2 degradation upon induction of lysosomal damage. HeLa Kyoto cells were transfected with non-coding siRNA or siRNA against STK38. 48 h post-transfection, cells were either mock- or LLOMe-treated and chased for the indicated time. (B) Quantification of (A). $n = 3$ biologically independent experiments. Two-way ANOVA with Sidak's multiple comparison test. Error bars represent the mean with SD.

