## [Peer Review File · The EMBO Journal]

ATXN3 regulates lysosome regeneration after damage by targeting K48-K63-branched ubiquitin chains

Maike Reinders, Bojana Kravic, Pinki Gahlot, Sandra Koska, Johannes van den Boom, Nina Schulze, Sophie Levantovsky, Stefan Kleine, Markus Kaiser, Yogesh Kulathu, Christian Behrends, and Hemmo Meyer

Corresponding author(s): Hemmo Meyer (Hemmo.Meyer@uni-due.de)

Review Timeline:

Submission Date:	26th Aug 24
Editorial Decision:	15th Oct 24
Revision Received:	10th Mar 25
Editorial Decision:	14th May 25
Revision Received:	7th Jul 25
Accepted:	11th Jul 25

Editor: Hartmut Vodermaier

Transaction Report:

Prof. Hemmo Meyer
Universität Duisburg-Essen
Faculty of Biology
Institute of Molecular Biology 1
Campus Essen
Essen 45141
Germany

15th Oct 2024

Re: EMBOJ-2024-118858
ATXN3 regulates lysosome regeneration after damage by targeting K48-K63-branched ubiquitin chains

Dear Hemmo,

Thank you for submitting your manuscript on ATXN3-mediated deubiquitination in lysosomal regeneration for our consideration. I am sorry for the delay in getting back to you with a response, owing mainly to busy travel schedules, slow referee pick-up, as well as delayed reports. We have now finally received feedback from three expert reviewers, copied below for your information. As you will see, the referees acknowledge the general experimental quality of the work, and appreciate that it further extends your previous work on this topic. At the same time, they raise various issues with the conclusiveness of the data and the overall depth of insight, which I feel would be important to address in order to make the study a compelling candidate for The EMBO Journal.

Should you be able to decisively address the key points noted in all three reports, we would be interested in pursuing a revised manuscript further for publication. I should remind you, however, that we only allow a single round of major revision, making it important to fully and carefully respond to each referee point at the time of resubmission. I would therefore encourage you to contact me with a revision plan and preliminary point-by-point response, or any other questions you may have in this regard, already during the early stages of your revision work, in order to discuss how key issues raised in the reports might best be resolved. We would also be open to extension of the regular three-months revision period if needed; our 'scooping protection' (meaning that competing work appearing elsewhere in the meantime will not affect our considerations of your study) would of course remain valid also throughout such an extension.

Further information on preparing, formatting and uploading a revised manuscript can be found below and in our Guide to Authors. Thank you again for the opportunity to consider this work for The EMBO Journal, and I look forward to hearing from you in due time.

With kind regards,

Hartmut

- size of the scale bars that are mandatory for all micrograph panels
- the statistical test used to generate error bars and P-values
- the type error bars (e.g., S.E.M., S.D.)
- the number (n) and nature (biological or technical replicate) of independent experiments underlying each data point
- Figures may not include error bars for experiments with $n < 3$; scatter plots showing individual data points should be used

instead.

9) To facilitate reproducibility and cross-laboratory adoption of methodologies, please structure the Materials & Methods section as outlined in our guide to authors, including a completed Reagents and Tools Table that can be downloaded from our author guidelines as well (<https://www.embopress.org/page/journal/14602075/authorguide#structuredmethods>).

10) Digital image enhancement is acceptable practice, as long as it accurately represents the original data and conforms to community standards. If a figure has been subjected to significant electronic manipulation, this must be clearly noted in the figure legend and/or the 'Materials and Methods' section. The editors reserve the right to request original versions of figures and the original images that were used to assemble the figure. Finally, we generally encourage uploading of numerical as well as gel/blot image source data; for details see: embopress.org/page/journal/14602075/authorguide#sourcedata

At EMBO Press, we ask authors to provide source data for the main manuscript figures. Our source data coordinator will contact you to discuss which figure panels we would need source data for and will also provide you with helpful tips on how to upload and organize the files.

Further information is available in our Guide For Authors:

In the interest of ensuring the conceptual advance provided by the work, we recommend submitting a revision within 3 months (13th Jan 2025). Please discuss the revision progress ahead of this time with the editor if you require more time to complete the revisions. Use the link below to submit your revision:

Link Not Available

Referee #1:

The cellular response to lysosomal damage involves mechanisms for membrane repair, regeneration, and lysophagy, though their coordination is not fully understood. This study reveals that the deubiquitinating enzyme ATXN3 is crucial for restoring lysosomal integrity after damage by targeting K48-K63-linked branched ubiquitin chains on regenerating lysosomes. Briefly, ATXN3 translocates to lysosomes after damage, colocalizing with its partner VCP/p97. Inactivating ATXN3 impairs the clearance of damaged lysosomes and hinders full lysosomal restoration. Mechanistically, ATXN3 and VCP/p97 remove K48-K63-branched ubiquitin conjugates on LAMP1-positive compartments, suggesting their role in lysosome regeneration. Overall, this is a clear and straightforward manuscript highlights ATXN3's key role in lysosomal function recovery and identifies K48-K63-branched ubiquitin chain-regulated regeneration as a critical aspect of the lysosomal damage response. Hence, this manuscript is appropriate for publication in the EMBO journal with some comments to improve the molecular details of the manuscript.

Main comments

1. The authors show that the PH-GFP sensor translocates to ATXN3-LAMP1 positive compartment after damage. Since the authors claim that ATXN3 regulates lysosome regeneration, is the recruitment of the PH-GFP sensor to regenerating lysosomes after damage affected by ATXN3 KO or siRNA?
2. The authors show accumulation of K48-K63 on damaged lysosomes. Do the K48-K63 chains colocalize also with PH-GFP-LAMP1 positive regenerating lysosome?
3. What are the ATXN4 substrates among the pool of K48-k63 ubiquitinated proteins on the lysosomes? Does this K48-K63 happens on specific proteins or is an unspecific signal?
4. The authors nicely demonstrated that knockout of ATXN3 or chemical inhibition of p97 led to an enhanced accumulation and persistence of K48-K63 branched chains. To strengthen their data, it is recommended that the authors conduct 1) a rescue experiment with ATXN3 wild type (WT) and C/A mutant, and 2) p97 knockout (KO) or siRNA experiments.
5. The expanded discussion on how deubiquitination of proteins on/in the lysosome can promote restoration of damaged lysosomes would be useful.

Minor comments:

1. In Figure 1B the authors show quantification and statistical analysis of Figure 1A. Is the difference between mock and time points 4, 6, 8 h statistically significant? Please show it in the figure.
2. The authors should provide a quantification for Fig. 1E and EV1A
3. Please provide standard deviation and statistical analysis in Fig.EV2B.
4. Please provide for Fig.3D, 4E, 5B, 5D, 6D, 6F the same quantification I asked in the second comment. All this quantification lacks a comparison with the control condition (undamaged) that is essential to understand if all the measured parameters are regulated by the damage, which is an important message of the paper.
5. The authors should provide a quantification for Fig 5E.

Referee #2:

Summary

The study entitled "ATXN3 regulates lysosome regeneration after damage by targeting K48-K63-branched ubiquitin chains" studied the involvement of ATXN3 in damaged lysosome clearance and its target is K48-K63-branched ubiquitin chains. Although the manuscript provides incremental findings from their previous papers, the following issues need to be better addressed:

Major concerns:

1. What is the substrate for ATXN3 on damaged lysosomes? Nanobody experiments showed the K48-K63-branched ubiquitin chains are removed, on which protein? Is the target the same as YOD1?
2. In the author's previous paper, YOD1(ELDR complex) is also required as DUB when lysosomes are damaged. Removal of K48 is required to proceed with lysophagy. Not only ATXN3 but also the ELDR complex is involved in the turnover of K48-K63-branched ubiquitin chains, as shown by the p97 inhibitor experiment. It seems p97 is involved in the turnover of K48 and K48-K63 branched ubiquitin chains. How are they linked? How does the DUB function of ATXN3 interlink with p97? What is the level of K48 and K63 alone when ATXN3 is KD/KO or CB-5083 in use? Why is the specific turnover of K48-K63-ub chains needed?
3. The authors claimed that K48 turnover is involved in phagophore formation and not K48-K63-Ub turnover. Less colocalization of LC3 with ATXN3 does not necessarily mean it is functioning at a later stage. Have the authors checked the EM to show that ATXN3 is not involved in phagophore formation? The authors need to show additional data to say that ATXN3 is involved in a later stage of phagophore formation.
4. Many pathways can recover lysosome damage: microautophagy, unconventional LC3-II, ESCRT, TFEB, etc. Data are

needed to confirm that ATXN3 is not required in other repair mechanisms but functions in later pathways.

Minor concerns:

1. Imaging data only shows several cells; thus, quantifications are required in all imaging data.
2. Merged data required both images (if 2 images were merged, 2 single images + merged image) for all. Some colocalization is challenging to see (Figure 1A, Figure 5A, E, Figure EV5); the authors should mark the colocalization.
3. Figure 6GH shows only nanobodies as evidence for ATXN3 and ELDR's turnover of K48-K63-ub chains. Additional biochemical data are needed, such as Lyso-IP damaged lysosomes and detection of the K48-K63-Ub chain.
4. In Figure 3A, is LLoMe added the whole time indicated? Is the unit min correct? In Skowyra ML. et al., Science 2018 paper, the recovery of lysosomes was observed in a matter of minutes after the washout.
5. It might relate to Major Concern 2; in Figure 6G, ATXN3 KO cells showed a similar level of accumulation of K48-K63-ub at both times after LLoMe was washed out; however, when an inhibitor of p97 was used (Figure 6I), the accumulated amount of K48-K63-Ub increased 8 hours after washout compared to 2 hours. What is the reason for this increment? LLoMe is removed, so no more damage is happening.
6. Endogenous localization of ATXN3 should be shown. If good antibodies are not available, perhaps knock in a tag to ensure that overexpression is not affecting the results. Western blot detection on Lyso-IP damaged lysosome fraction might also be ok.
7. The PH domain is there but not repaired; if so, why does that show "in the process of repairing?"

Referee #3:

In the manuscript "ATXN3 regulates lysosome regeneration after damage by targeting K48-K63-branched ubiquitin chains", Reinders et al identified deubiquitinating enzyme ATXN3 as a novel regulator of lysosomal regeneration after damage. ATXN3 localizes to non-acidic regenerating lysosomes during lysosomal damage. Cooperating with p97, ATXN3 targets damage-induced K48-K63 branched conjugates on regenerating lysosomes and turn over them. This process regulates restoration of lysosomal capacity, and it is a critical element of the lysosomal damage response.

Most of the data in this manuscript are convincing and presented by well-designed experiments. I have two major concerns regarding the main message of the manuscript. One is that the authors claim that ATXN3 is not involved in phagophore formation, but more convincing data is required to demonstrate this. The authors use LC3 as a marker for phagophore formation, but it is known that non-canonical autophagy occurs upon LLOMe treatment. Therefore, there is still concern about whether ATXN3 is involved in phagophore formation during lysosomal damage response. The other concern is that it is not clear whether ATXN3 functions in regeneration of lysosomes. The authors clearly show that ATXN3 localizes to non-acidified regenerating lysosomes, but there is no evidence that ATXN3 functions in lysosome regeneration. It is also not clear how turnover of K48-K63 branched ubiquitin conjugates is involved in regeneration of lysosomes. Addressing these concerns would strengthen the conclusion of the manuscript.

Major points:

1. The authors claim that ATXN3 is not involved in early lysosomal repair pathways such as ESCRT-mediated membrane invagination, but only showed the localization of IST1 and ATXN3. It is necessary to show recruitment of ESCRTs (e.g ALIX, IST1) to damaged lysosomes in ATXN3 KO cells.
2. The authors claim that ATXN3 is not involved in phagophore formation based on the findings that depletion of ATXN3 did not affect recruitment of p62 or LC3. However, it is known that non-canonical LC3 lipidation occurs on lysosomes upon LLOMe treatment. Therefore, other ATGs, such as FIP200, should be used as a marker for phagophore formation, rather than LC3.
3. As the authors mentioned in the manuscript, previous studies have shown that ATXN3 regulates phagophore formation. In addition, the TMEM192-mKeima assay (Fig.4A) shows that lysophagy is suppressed in ATXN3 KO cells. Did the authors examine if deletion of ATXN3 affects phagophore formation upon starvation? Even though ATXN3 is required for lysosome regeneration, re-acidification of lysosomes after LLOMe treatment is normal in ATXN3 KO cells (Fig.3B), so there must be functional lysosomes in ATXN3 KO cells. Therefore, there are still concerns about whether phagophore formation is normal in ATXN3 KO cells.
4. In Fig.4E, the authors state that "ATXN3-GFP only partially colocalized with LC3", but this may be because the number of LC3 dots is lower than ATXN3. When comparing LC3+ATXN3+LAMP1+ and LC3+ATXN3-LAMP1+, it appears that half of the LC3 dots colocalize with ATXN3.

5. In Fig.5A, the images of LAMP1 are not clear, so please replace them.
6. The authors clearly showed that ATXN3 localized to non-acidified regenerating lysosomes, but there is no evidence that ATXN3 functions in lysosome regeneration. How is turnover of K48-K63 branched ubiquitin conjugates involved in regeneration of lysosomes?
7. Is the number of PH+LAMP+ dots affected by deletion of ATXN3?
8. Why did chemical inhibition of p97 cause a much stronger accumulation of K48-K63-branched chain than knockout of ATXN3? Other deubiquitinating enzymes such as YOD1 are involved?
9. According to Fig.5C, Fig.6C, most of p97 seems to localize to non-acidified regenerating lysosomes with ATXN3, indicating that p97 mainly functions lysosome regeneration. On the other hand, the authors have previously reported that p97 is required for LC3 recruitment and phagophore formation by removing K48 or ubiquitylated CNN2. Do these also occur on non-acidified regenerating lysosomes? It would be better to discuss the relevance of this work to the previous reports in the discussion section.

Point-by-point response EMBOJ-2024-118858R

Preface:

We thank the referees for their generally positive feedback. We believe we have, based on the input, significantly improved the manuscript. As some of suggestions overlapped between the referees, we would like to first briefly summarize the major improvements:

Most importantly, we applied a proteomics approach on nanobody isolates to identify substrates modified with K48-K63 chains. We focused on LAMP2 because LAMP2 has previously been shown to be ubiquitylated and degraded upon lysosomal damage (Yoshida et al., 2017). We now show that LAMP2 is degraded dependent on ATXN3 during the lysosomal damage response, thereby revealing a molecular role for ATXN3 and branched chains. Based on sensitivity to Bafilomycin A, timing and association with the regulator STK38, we discuss that proteins such as LAMP2 are degraded by micro-lysophagy to promote regeneration of a subset of lysosomes.

We demonstrate biochemically that, in ATXN3-compromised cells, LC3 lipidation is not reduced but that, instead, autophagy flux is impaired, consistent with our conclusion that lysophagy is affected at a late stage. We also show that damaged-induced TECPR1 or ALIX recruitment is not affected in ATXN3-compromised cells.

We directly compare ATXN3 function with that of the other p97-associated DUB, YOD1, and find that its function differs regarding K48-K63-branch chain processing and the effect on autophagic flux. We discuss that ATXN3 has an additional function distinct from the previously described role of p97 and YOD1 in phagophore formation. This is in line with the established notion that p97 is a multifunctional enzyme whose activity is differently engaged by distinct cofactor proteins.

In addition, we added further controls, biochemical validation, statistics and split channels as requested. We also now demonstrate that a p97-binding deficient mutant of ATXN3 (VBM) cannot the Gal3 clearance defect in ATXN3 KO cells demonstrating that ATXN3 cooperates with p97. In doing so, we noted that we previously included a wrong micrograph for the C14A mutant which we now corrected. The effect is identical, so this does not affect the results or conclusions.*

Please see below for the detailed point-by-point response.

Referee #1:

The cellular response to lysosomal damage involves mechanisms for membrane repair, regeneration, and lysophagy, though their coordination is not fully understood. This study reveals that the deubiquitinating enzyme ATXN3 is crucial for restoring lysosomal integrity after damage by targeting K48-K63-linked branched ubiquitin chains on regenerating lysosomes. Briefly, ATXN3 translocates to lysosomes after damage, colocalizing with its partner VCP/p97. Inactivating ATXN3 impairs the clearance of damaged lysosomes and hinders full lysosomal restoration. Mechanistically, ATXN3 and VCP/p97 remove K48-K63-branched ubiquitin conjugates on LAMP1-positive compartments, suggesting their role in lysosome regeneration. Overall, this is a clear and straightforward manuscript highlights ATXN3's key role in lysosomal function recovery and identifies K48-K63-branched ubiquitin

chain-regulated regeneration as a critical aspect of the lysosomal damage response. Hence, this manuscript is appropriate for publication in the EMBO journal with some comments to improve the molecular details of the manuscript.

Main comments

1. The authors shows that the PH-GFP sensor translocates to ATXN3-LAMP1 positive compartment after damage. Since the authors claim that ATXN3 regulates lysosome regeneration, is the recruitment of the PH-GFP sensor to regenerating lysosomes after damage affected by ATXN3 KO or siRNA?

We could not detect that difference in our experiments which is probably due to the long duration of the recovery. We now provide the DQ-BSA assay (for proteolytic capacity) in the main figure 4 showing that recovery is delayed in ATXN3 KO even after 20 h. Thus, ATXN3 affects lysosome regeneration.

2. The authors show accumulation of K48-K63 on damaged lysosomes. Do the K48-K63 chains colocalize also with PH-GFP-LAMP1 positive regenerating lysosome?

As requested, we now provide these data showing that the majority of K48-K63-decorated lysosomes are positive for PH-GFP (new Fig. EV4EF)

3. What are the ATXN4 substrates among the pool of K48-k63 ubiquitinated proteins on the lysosomes? Does this K48-K63 happens on specific proteins or is an unspecific signal?

To address this question, we have engaged in a major effort (new Fig. 6). As outlined in the preface to this response, we now show that LAMP2 is modified with K48-K63 chains and that this regulates degradation of a fraction of LAMP2 after lysosomal damage, most likely to eliminate denatured proteins during membrane perturbations. Moreover, we provide evidence consistent with this degradation being mediated by micro-autophagy as part of the lysosomal regeneration.

4. The authors nicely demonstrated that knockout of ATXN3 or chemical inhibition of p97 led to an enhanced accumulation and persistence of K48-K63 branched chains. To strengthen their data, it is recommended that the authors conduct 1) a rescue experiment with ATXN3 wild type (WT) and C/A mutant, and 2) p97 knockout (KO) or siRNA experiments.

We now provide biochemical evidence for turnover of branched chains by ATXN3 during lysosomal damage (Fig. 5GH). Moreover, as requested, we now confirm the effect on K48-K63 dynamics by siRNA (Fig. EV4IJ). We performed extensive ATXN3 rescue experiments for the Gal3 clearance assays. The rescue experiments are difficult because ATXN3 wt overexpression can be deleterious and needs to be carefully fine-tuned, which we could not prioritize in this revision. However, we feel that the phenocopies of ATXN3 KO, ATXN3 siRNA and p97 inhibition conclusively demonstrate our point. The p97 inhibitor experiments are the approach of choice because CB-5073 is highly characterized, whereas p97 KO are lethal and p97 siRNA causes broad cytopathic effects.

5. The expanded discussion on how deubiquitination of proteins on/in the lysosome can promote restoration of damaged lysosomes would be useful.

We have now expanded the discussion. Based on the new data, we speculate that branched ubiquitin chains regulate degradation of a fraction of LAMP2 (that may be denatured during the damage) and that this degradation is mediated by micro-lysophagy.

Minor comments:

1. In Figure 1B the authors show quantification and statistical analysis of Figure 1A. Is the difference between mock and time points 4, 6, 8 h statistically significant? Please show it in the figure.

As requested, we tested the difference and now show that it is significant. For space reasons, the values are listed in the figure legend.

2. The authors should provide a quantification for Fig. 1E and EV1A

As requested, we now provide quantification of ATXN3 localization in ARPE19 (former Fig. 1E). We removed Fig. EV1A (Terfenadine-induced damage). While the lysosomotropic terfenadine induces lysosome damage (leading to consistent recruitment of ATXN3), the events are too rare for meaningful automatic quantification with our current setup. Nevertheless, we provide evidence for relevance in LLOMe and laser-induced lipid peroxidation as two very different types of membrane perturbations.

3. Please provide standard deviation and statistical analysis in Fig. EV2B.

This experiment compares different ATXN3 KO clones representing already the replica. In Fig. 2, we analyzed one clone in detail with statistics and rescue setups.

4. Please provide for Fig. 3D, 4E, 5B, 5D, 6D, 6F the same quantification I asked in the second comment. All this quantification lacks a comparison with the control condition (undamaged) that is essential to understand if all the measured parameters are regulated by the damage, which is an important message of the paper.

We added the requested P-values for Fig. 1AB (comment 1) and former Fig. 6AB (now 5AB, K48-K63 chain localization), as well as many other figures. For space reasons, we put them in part in the figure legends. In the other figures listed by the referee, we compare subpopulations of lysosomes with LLOMe-induced localizations of different markers. The induction of translocation (mock versus LLOMe) was quantified and tested with p values in previous experiments already. For example, in Fig. EV2D (formerly Fig. 3D) that compares how many ATXN3+ lysosomes are also IST+, we showed significance of ATXN3 translocation (mock versus LLOMe) already in Fig. 1AB.

5. The authors should provide a quantification for Fig 5E.

This concerns the costaining of GFP-PH with LysoTracker by live-cell super-resolution microscopy. As this microscopy is very time-consuming and access limited, we decided to remove this data set because we could not acquire enough data for statistics as requested due to prioritization of further functional exploration during the limited time of the revision. In any case, old Fig. 5E only confirmed what can be concluded of 5A (lack of colocalization of ATXN3 and LysoTracker) and 5C (colocalization of ATXN3 and GFP-PH). Moreover, we now provide colocalization of K48-K63 chains and GFP-PH including quantification.

Referee #2:

Summary

The study entitled "ATXN3 regulates lysosome regeneration after damage by targeting K48-K63-branched ubiquitin chains" studied the involvement of ATXN3 in damaged lysosome clearance and its target is K48-K63-branched ubiquitin chains. Although the manuscript provides incremental findings from their previous papers, the following issues need to be better addressed:

Major concerns:

1. What is the substrate for ATXN3 on damaged lysosomes? Nanobody experiments showed the K48-K63-branched ubiquitin chains are removed, on which protein? Is the target the same as YOD1?

To address this point, we first confirmed biochemically that K48-K63 conjugates accumulate in ATXN3 KO and show that this does not occur in YOD1 KO cells (Fig. 5G, EV4K). We then performed mass spec on K48-K63-conjugates isolated specifically with the nanobody after lysosomal damage (new Fig. 6). We identified several targets including LAMP2. We further showed that LAMP2 is degraded in the lysosome after damage dependent on ATXN3. This provides a molecular explanation for the role of ATXN3 and branched ubiquitin chains in the lysosome damage response. In addition, given the observed association with micro-autophagy regulator STK38, the sensitivity to Bafilomycin, the specificity compared to LAMP1 and the relative early timing compared to overall lysophagy, we discuss that LAMP2 may be degraded by micro-autophagy that helps regenerate lysosomes by removing denatured proteins and affected sections of the limiting membrane.

2. In the author's previous paper, YOD1(ELDR complex) is also required as DUB when lysosomes are damaged. Removal of K48 is required to proceed with lysophagy. Not only ATXN3 but also the ELDR complex is involved in the turnover of K48-K63-branched ubiquitin chains, as shown by the p97 inhibitor experiment. It seems p97 is involved in the turnover of K48 and K48-K63 branched ubiquitin chains. How are they linked? How does the DUB function of ATXN3 interlink with p97? What is the level of K48 and K63 alone when ATXN3 is KD/KO or CB-5083 in use? Why is the specific turnover of K48-K63-ub chains needed?

We see that this may cause confusion. Of note, p97 is an abundant enzyme that is engaged in different functions by distinct cofactor proteins. We have now tested the other p97-associated DUB, YOD1, and show that it (in contrast to ATXN3) is not involved in K48-K63 processing (Fig. EV4KL) and does not lead to a late-stage inhibition of lysophagic flux (Fig. EV3DE). YOD1 has therefore distinct function in processing K48-linked ubiquitin conjugates as previously described. However, technically, we cannot discriminate K48-linked conjugates from K48-K63-branched conjugates, because the K48-specific antibodies will stain K48-linked and K48-K63-branched chains. Importantly, the new work added to the manuscript now links K48-K63-branched chains to LAMP2 degradation likely through micro-autophagy (Fig. 6).

3. The authors claimed that K48 turnover is involved in phagophore formation and not K48-K63-Ub turnover. Less colocalization of LC3 with ATXN3 does not necessarily mean it is functioning at a later stage. Have the authors checked the EM to show that ATXN3 is not involved in phagophore formation? The authors need to show additional data to say that ATXN3 is involved in a later stage of phagophore formation.

We show that inactivation of ATXN3 does not lead to p62 and LC3 recruitment to terminally damaged lysosomes. We now support this by analyzing the LC3 lipidation status at different time points. We see an accumulation of lipidated LC3 rather than on actual lipidation (Fig. 3DE). The phenotype is therefore distinct from other p97-associated factors. Importantly, we reveal a positive function of

ATXN3 in that ATXN3 regulates degradation of a fraction of LAMP2 specifically, and that LAMP2 is modified with K48-K63-branched chains (Fig. 6).

4. Many pathways can recover lysosome damage: microautophagy, unconventional LC3-II, ESCRT, TFEB, etc. Data are needed to confirm that ATXN3 is not required in other repair mechanisms but functions in later pathways.

We agree that many additional pathways contribute to lysosomal recovery. Moreover, we do not exclude that ATXN3 regulates several processes, although we now (in response to this point) evaluated TECPR1, IST1 and ALIX recruitment and found no changes in ATXN3 KO cells. Importantly however, we now demonstrate that ATXN3 promotes LAMP2 degradation and that this is associated with micro-autophagy. We discuss that micro-autophagy may be regulated by K48-K63-branched chains and mediate removal of proteins like LAMP2 and sections of the limiting membrane for lysosomal recovery.

Minor concerns:

1. Imaging data only shows several cells; thus, quantifications are required in all imaging data.

As requested, we have included new quantifications to many figures. In two cases, we could not prioritize additional replica and have therefore removed the data altogether (PH-GFP/LysoTracker colocalisation by SIM; Terfenadine-induced damage) as they are not essential for our conclusions.

2. Merged data required both images (if 2 images were merged, 2 single images + merged image) for all. Some colocalization is challenging to see (Figure 1A, Figure 5A, E, Figure EV5); the authors should mark the colocalisation.

As requested, we have added split channels in the listed figure and elsewhere. We now show blow-up insets to emphasize the colocalisation.

3. Figure 6GH shows only nanobodies as evidence for ATXN3 and ELDR's turnover of K48-K63-ub chains. Additional biochemical data are needed, such as Lyso-IP damaged lysosomes and detection of the K48-K63-Ub chain.

As requested, we now provide the biochemical evidence (nanobody pulldown analyzed by Western blotting), supporting our conclusions.

4. In Figure 3A, is LLoMe added the whole time indicated? Is the unit min correct? In Skowyra ML. et al., Science 2018 paper, the recovery of lysosomes was observed in a matter of minutes after the washout.

Agreed. We followed a protocol of the Stenmark group without washout of LLOMe (Radulovic et al., 2018), which delays repair.

5. It might relate to Major Concern 2; in Figure 6G, ATXN3 KO cells showed a similar level of accumulation of K48-K63-ub at both times after LLoMe was washed out; however, when an inhibitor of p97 was used (Figure 6I), the accumulated amount of K48-K63-Ub increased 8 hours after washout compared to 2 hours. What is the reason for this increment? LLoMe is removed, so no more damage is happening.

Like for many regulatory processes, the level of K48-K63-branched ubiquitin chains is the result of ongoing dynamic formation and turnover as a response of the previous damage insult. Tampering with the turnover can therefore lead to increased steady state levels.

6. Endogenous localization of ATXN3 should be shown. If good antibodies are not available, perhaps knock in a tag to ensure that overexpression is not affecting the results. Western blot detection on Lyso-IP damaged lysosome fraction might also be ok.

As requested, we now show translocation of endogenous ATXN3 with a commercial antibody. Specificity of the staining is demonstrated in ATXN3 KO cells.

7. The PH domain is there but not repaired; if so, why does that show "in the process of repairing?"

PI(4,5)P2 is generated on damaged lysosomes and associated with the process of lysosome regeneration (Bhattacharya et al, 2023). We could not follow the PH-GFP pattern over a long time but now show (using the DQ-BSA assay) in main figure 4EF that lysosomes still have not fully recovered after 20 h and this is further delayed in ATXN3 KO cells.

Referee #3:

In the manuscript "ATXN3 regulates lysosome regeneration after damage by targeting K48-K63-branched ubiquitin chains", Reinders et al identified deubiquitinating enzyme ATXN3 as a novel regulator of lysosomal regeneration after damage. ATXN3 localizes to non-acidic regenerating lysosomes during lysosomal damage. Cooperating with p97, ATXN3 targets damage-induced K48-K63 branched conjugates on regenerating lysosomes and turn over them. This process regulates restoration of lysosomal capacity, and it is a critical element of the lysosomal damage response.

Most of the data in this manuscript are convincing and presented by well-designed experiments. I have two major concerns regarding the main message of the manuscript. One is that the authors claim that ATXN3 is not involved in phagophore formation, but more convincing data is required to demonstrate this. The authors use LC3 as a marker for phagophore formation, but it is known that non-canonical autophagy occurs upon LLOMe treatment. Therefore, there is still concern about whether ATXN3 is involved in phagophore formation during lysosomal damage response. The other concern is that it is not clear whether ATXN3 functions in regeneration of lysosomes. The authors clearly show that ATXN3 localizes to non-acidified regenerating lysosomes, but there is no evidence that ATXN3 functions in lysosome regeneration. It is also not clear how turnover of K48-K63 branched ubiquitin conjugates is involved in regeneration of lysosomes. Addressing these concerns would strengthen the conclusion of the manuscript.

Major points:

1. The authors claim that ATXN3 is not involved in early lysosomal repair pathways such as ESCRT-mediated membrane invagination, but only showed the localization of IST1 and ATXN3. It is necessary to show recruitment of ESCRTs (e.g ALIX, IST1) to damaged lysosomes in ATXN3 KO cells.

As requested, we now show recruitment of IST1 and also ALIX. We find no effect of ATXN3 KO consistent with our conclusions.

2. The authors claim that ATXN3 is not involved in phagophore formation based on the findings that depletion of ATXN3 did not affect recruitment of p62 or LC3. However, it is known that non-canonical LC3 lipidation occurs on lysosomes upon LLOMe treatment. Therefore, other ATGs, such as FIP200, should be used as a marker for phagophore formation, rather than LC3.

As requested, we analyzed the non-canonical LC3 lipidation of the single (lysosomal) membrane, which occurs early after damage. Moreover, we stained TECPR1 which mediates much of the non-canonical LC3 lipidation. Neither of the two are affected in ATXN3 KO cells.

3. As the authors mentioned in the manuscript, previous studies have shown that ATXN3 regulates phagophore formation. In addition, the TMEM192-mKeima assay (Fig.4A) shows that lysophagy is suppressed in ATXN3 KO cells. Did the authors examine if deletion of ATXN3 affects phagophore formation upon starvation? Even though ATXN3 is required for lysosome regeneration, re-acidification of lysosomes after LLOMe treatment is normal in ATXN3 KO cells (Fig.3B), so there must be functional lysosomes in ATXN3 KO cells. Therefore, there are still concerns about whether phagophore formation is normal in ATXN3 KO cells.

We now provide Western blot analysis of LC3-lipidation showing that LC3 lipidation occurs normally and that autophagy flux is affected consistent with the mKeima assay. Because this is stress-induced selective autophagy, we do not make statements about starvation induced autophagy. We find a rather large fraction of damaged lysosomes associated with ATXN3 and show that degradative

capacity of the lysosomal system (DQ-BSA assay) is significantly affected in ATXN3 KO cells, which therefore may account for reduced lysophagy.

4. In Fig.4E, the authors state that "ATXN3-GFP only partially colocalized with LC3", but this may be because the number of LC3 dots is lower than ATXN3. When comparing LC3+ATXN3+LAMP1+ and LC3+ATXN3-LAMP1+, it appears that half of the LC3 dots colocalize with ATXN3.

While there is some overlap, the majority of ATXN3+ lysosomes are negative for LC3, suggesting that ATXN3 does not regulate LC3 but there fulfils a different function, which we now further support in our LC3 lipidation analysis.

5. In Fig.5A, the images of LAMP1 are not clear, so please replace them.

What may appear to be "not clear" is the result of the inherent image processing associated with structure illumination microscopy (SIM). The background pattern stems from the ATXN3 remaining diffuse in the cytosol. We feel the images are state-of-the-art super-resolution images.

6. The authors clearly showed that ATXN3 localized to non-acidified regenerating lysosomes, but there is no evidence that ATXN3 functions in lysosome regeneration. How is turnover of K48-K63 branched ubiquitin conjugates involved in regeneration of lysosomes?

We now show in the main Fig. 4EF that lysosome recovery is delayed in ATXN3 KO cells as shown with the DQ-BSA assay. With regard to substrates, we provide evidence that ATXN3 regulates degradation of a fraction of LAMP2 that is modified with K48-K63-branched ubiquitin chains. Furthermore, we provide evidence that this is linked to the recently described micro-autophagy. We discuss that micro-autophagy helps regeneration by eliminating parts of limiting membrane and proteins that are possibly denatured during membrane damage.

7. Is the number of PH+LAMP+ dots affected by deletion of ATXN3?

We could not detect a difference here but see a reduction in lysosome recovery with the DQ-BSA assay (Fig. 4EF) in ATXN3 KO cells demonstrating a defect in lysosome recovery.

8. Why did chemical inhibition of p97 cause a much stronger accumulation of K48-K63-branched chain than knockout of ATXN3? Other deubiquitinating enzymes such as YOD1 are involved?

A dominant effect of p97 inhibition was already observed in the paper linking p97/ATXN3 to K48-K63 chains (Lange et al., 2024). We agree that other DUBs may help (when ATXN3 is compromised) although we now show that YOD1 KO alone does not lead to branched chain accumulation.

9. According to Fig.5C, Fig.6C, most of p97 seems to localize to non-acidified regenerating lysosomes with ATXN3, indicating that p97 mainly functions lysosome regeneration. On the other hand, the authors have previously reported that p97 is required for LC3 recruitment and phagophore formation by removing K48 or ubiquitylated CNN2. Do these also occur on non-acidified regenerating lysosomes? It would be better to discuss the relevance of this work to the previous reports in the discussion section.

As requested, we are now elaborating on this issue in the discussion section.

Prof. Hemmo Meyer
Universität Duisburg-Essen
Faculty of Biology
Universitaetsstr. 2
Essen 45141
Germany

14th May 2025

Re: EMBOJ-2024-118858R
ATXN3 regulates lysosome regeneration after damage by targeting K48-K63-branched ubiquitin chains

Dear Hemmo,

Thank you again for submitting your revised manuscript to The EMBO Journal, and my sincere apologies for the delay in getting back to you with the outcome of its re-review. As you will see, the referees remain at this stage still divided about the study, and this has necessitated further discussions both within the editorial team and with the referees. While referee 1 was for the most part satisfied with the revisions and responses, referees 2 and 3 both retain important reservations, in particular regarding the evidence supporting ATXN3 functioning at a late stage of phagophore formation. During our follow-up cross-consultations, also referee 1 agreed that it would be helpful to obtain additional data to better distinguish early from late autophagy steps, as requested by the other referees.

We normally allow only a single round of major revision - to avoid repeated rounds of partial improvements and to give authors a decisive commitment from our side. However, given that you have already made significant efforts and addressed a majority of issues during the first revision, I would in this case allow an exceptional second round of experimental revision, to allow you to clarify the remaining open points. While I would not expect in-depth follow-up on every single point raised by referee 3, the key issue would be decisively determining the stage at which ATXN3 functions, by using a more direct marker such as suggested by referee 2 or by referee 3 (point 1). Furthermore, it would clearly be helpful if you could add some data and good argumentation in response to points 3 and 4 of referee 3. With such additional strengthening of the study, we would be happy to eventually accept it for EMBO Journal publication. As always, I would be open to considering possible options for revising based on a preliminary response letter, which I might in this case discuss with the critical referee(s) to get their views on how promising they would seem.

I am therefore returning the manuscript to you for a second, final round of experimental revision, with the link below for eventual resubmission. When preparing a re-revised manuscript addressing the persistent scientific issues, please also take care of the following editorial points that we had noted during our routine revision checks:

- Please carefully go through the reference list and make sure that each reference is complete with citation year, volume, and page/locator numbers.
- Our routine pre-acceptance image checks indicated that certain microscopy panels, or overlapping fields, are repeatedly displayed between Figures 3B and EV3B. This needs to be rationalized and clearly stated in all respective figure legends, or where necessary addressed via replacement figure panels.

Once again, sorry for the delay with this re-review process, and I look forward to hearing back from you in due time.

With kind regards,

Hartmut

*** PLEASE NOTE: All revised manuscripts are subject to initial checks for completeness and adherence to our formatting guidelines. Revisions may be returned to the authors and delayed in their editorial re-evaluation if they fail to comply with the following requirements (see also our Guide to Authors for further information):

- 1) Every manuscript requires a Data Availability section (even if only stating that no deposited datasets are included). Primary

datasets or computer code produced in the current study have to be deposited in appropriate public repositories prior to resubmission, and reviewer access details provided in case that public access is not yet allowed. Further information: embopress.org/page/journal/14602075/authorguide#dataavailability

9) To facilitate reproducibility and cross-laboratory adoption of methodologies, please structure the Materials & Methods section as outlined in our guide to authors, including a completed Reagents and Tools Table that can be downloaded from our author guidelines as well (<https://www.embopress.org/page/journal/14602075/authorguide#structuredmethods>).

10) Digital image enhancement is acceptable practice, as long as it accurately represents the original data and conforms to community standards. If a figure has been subjected to significant electronic manipulation, this must be clearly noted in the figure legend and/or the 'Materials and Methods' section. The editors reserve the right to request original versions of figures and the original images that were used to assemble the figure. Finally, we generally encourage uploading of numerical as well as gel/blot image source data; for details see: embopress.org/page/journal/14602075/authorguide#sourcedata

In the interest of ensuring the conceptual advance provided by the work, we recommend submitting a revision within 3 months (12th Aug 2025). Please discuss the revision progress ahead of this time with the editor if you require more time to complete the revisions. Use the link below to submit your revision:

Link Not Available

Referee #1:

The authors have successfully addressed the comments from the first review. They have provided several novel evidences for

the role of ATXN3 in the later stages of lysophagy including that LC3 lipidation is not reduced while autophagy flux is, and that damaged induced TECPR1 or ALIX recruitment is not affected in ATXN3 deficient cells. No further comments.

Referee #2:

In this revised manuscript, the authors have addressed most of the concerns raised by referees, except for one: demonstrating that ATXN3 functions at a later stage. Two referees requested additional evidence beyond LC3-II. To distinguish between the early and late stages of phagophore formation, the best ATG marker to use is the Atg12-Atg5-Atg16 complex. This should be addressed.

Referee #3:

Although the authors have made significant efforts to address the concerns raised in the previous review, including attempts to identify the target of K48-K63 branched chains and to elucidate the significance of its turnover, a few important concerns remain, as outlined below.

1. The data presented to support the claim that ATXN3 is not involved in phagophore formation remain insufficient. The authors use LC3 lipidation as a marker; however, under LLOMe treatment, this marker is not appropriate due to the induction of non-canonical LC3 lipidation. In such conditions, canonical and non-canonical lipidation cannot be distinguished, and non-canonical processes are likely predominant [PMID: 32989250]. Indeed, in Figure EV3, LC3 lipidation persists in YOD1 KO cells even when phagophore formation is suppressed, further supporting the idea that non-canonical LC3 lipidation is predominant in this context. To properly assess whether phagophore formation is unaffected in ATXN3 KO cells, it is important to use a more direct marker such as FIP200. Although this point was raised in the previous review, the experiments directly assessing phagophore formation using FIP200 have not been included. As also noted by Reviewer #2, complementary confirmation by electron microscopy would greatly strengthen the authors' conclusion; however, this request also appears to have been left unaddressed in the current revision. These two approaches are essential for addressing this point.

2. While the ubiquitination of LAMP2 is evident from both previous studies and the present work, the use of a pan-ubiquitin antibody does not allow determination of whether LAMP2 is specifically modified by K48-K63 branched chains.

3. The paper cited by the authors indicates that LAMP2 is degraded through lysophagy. Therefore, the accumulation of LAMP2 in ATXN3 KO cells could potentially be explained by impaired lysophagy alone, without the need to invoke microautophagy or K48-K63 branched chain modification. The authors also refer to the difference in degradation between LAMP1 and LAMP2 as evidence for selective microautophagy; however, the observed differences appear to be relatively minor, making this argument less convincing. Furthermore, the claim that LAMP2 is degraded via microautophagy based solely on its colocalization with STK38 seems overly speculative. A more direct assessment of microautophagy activity, such as EGFP-TRPML1 cleavage assay [PMID:32916093] in ATXN3 KO cells and in rescue experiments using the C14D mutant, would provide stronger support for this interpretation.

4. STK38 has been shown to be required for lysosomal membrane repair mediated by the ESCRT machinery, an early response to lysosomal damage [PBID: 37987447]. In contrast, the authors demonstrate that ATXN3 is recruited to damaged lysosomes at a later stage and is not involved in ESCRT-mediated repair. Therefore, the proposed model-in which ATXN3 promotes the degradation of a fraction of LAMP2 via turnover of its K48-K63-branched ubiquitin conjugates through STK38-mediated microautophagy-appears inconsistent in both timing and mechanism. For these reasons, outlined in points #2, #3, and #4, the interpretation of Figure 6 remains unconvincing.

Point-by-point response

Referee #1:

The authors have successfully addressed the comments from the first review. They have provided several novel evidences for the role of ATXN3 in the later stages of lysophage including that LC3 lipidation is not reduced while autophagy flux is, and that damaged induced TECPR1 or ALIX recruitment is not affected in ATXN3 deficient cells. No further comments.

Referee #2:

In this revised manuscript, the authors have addressed most of the concerns raised by referees, except for one: demonstrating that ATXN3 functions at a later stage. Two referees requested additional evidence beyond LC3-II. To distinguish between the early and late stages of phagophore formation, the best ATG marker to use is the Atg12-Atg5-Atg16 complex. This should be addressed.

Because the ATG16 complex mediates both canonical (phagophore-associated) and non-canonical LC3 lipidation, we instead opted for ULK1 that exclusively regulates phagophore formation. ULK1 was also used by Ogura et al., 2023, that showed the role of STK38 in microautophagy and excluded macroautophagy. We now show by microscopy that ULK1 recruitment is not affected by compromised ATXN3 function (EV3 H and I). Consistent with that, we now also show in Western blots that inhibition of ULK1/2 and PI3KC3 reduces damage-associated LC3 lipidation (EV3 F and G), as expected, demonstrating that a large fraction of lipidated LC3 after lysosome damage is associated is phagophore-associated. In contrast, ATXN3 KO does not have this effect on LC3-II levels. We conclude that ATXN3 is not involved in phagophore-associated LC3 lipidation.

Referee #3:

Although the authors have made significant efforts to address the concerns raised in the previous review, including attempts to identify the target of K48-K63 branched chains and to elucidate the significance of its turnover, a few important concerns remain, as outlined below.

1. The data presented to support the claim that ATXN3 is not involved in phagophore formation remain insufficient. The authors use LC3 lipidation as a marker; however, under LLOMe treatment, this marker is not appropriate due to the induction of non-canonical LC3 lipidation. In such conditions, canonical and non-canonical lipidation cannot be distinguished, and non-canonical processes are likely predominant [PMID: 32989250].

Indeed, in Figure EV3, LC3 lipidation persists in YOD1 KO cells even when phagophore formation is suppressed, further supporting the idea that non-canonical LC3 lipidation is predominant in this context. To properly assess whether phagophore formation is unaffected in ATXN3 KO cells, it is important to use a more direct marker such as FIP200. Although this point was raised in the previous review, the experiments directly assessing phagophore formation using FIP200 have not been included. As also noted by Reviewer #2, complementary confirmation by electron microscopy would greatly strengthen the authors' conclusion; however, this request also appears to have been left unaddressed in the current revision. These two approaches are essential for addressing this point.

We now show that inhibition of ULK1/2 and PI3KC3 reduces damage-induced LC3 lipidation significantly. This demonstrates that canonical, phagophore-associated LC3 lipidation is a detectable fraction in the process. We also show that this fraction is not affected by ATXN3 KO. The significance of phagophore-associated LC3 lipidation is also demonstrated by microscopy data provided for the reviewer (see below).

Figure for referees not shown.

ULK1/2 and PI3KC3 inhibition largely reduces lysosomes associated LC3 (which we do not observe with compromised ATXN3 function shown in the manuscript).

As requested, we now also show that recruitment of the ULK1 complex is not affected by ATXN3 KO (EV3 H and I). We opted for direct detection of ULK1 rather than its partner, FIP200. Of note, ATXN3 KO thus phenocopies STK38 depletion that also does not affect ULK1 recruitment (Ogura et al., 2023).

2. While the ubiquitination of LAMP2 is evident from both previous studies and the present work, the use of a pan-ubiquitin antibody does not allow determination of whether LAMP2 is specifically modified by K48-K63 branched chains.

Branched chains cannot technically be detected on Western blot with the nano-body. Hence, we had to use pan-ubiquitin antibody. Nevertheless, LAMP2 was detected prominently in our MS screen in branched-chain isolates. Moreover, we show that LAMP2 is regulated by ATXN3 (and STK38) and ATXN3 regulates branched chains.

3. The paper cited by the authors indicates that LAMP2 is degraded through lysophagy.

Therefore, the accumulation of LAMP2 in ATXN3 KO cells could potentially be explained by impaired lysophagy alone, without the need to invoke microautophagy or K48-K63 branched chain modification. The authors also refer to the difference in degradation between LAMP1 and LAMP2 as evidence for selective microautophagy; however, the observed differences appear to be relatively minor, making this argument less convincing. Furthermore, the claim that LAMP2 is degraded via microautophagy based solely on its colocalization with STK38 seems overly speculative. A more direct assessment of microautophagy activity, such as EGFP-TRPML1 cleavage assay [PMID:32916093] in ATXN3 KO cells and in rescue experiments using the C14D mutant, would provide stronger support for this interpretation.

The experiments in the cited paper (Yoshida et al., 2017) showed that LAMP2 is degraded by autophagy/lysophagy but could not discriminate whether this is through macro-autophagy or micro-autophagy. In fact, micro-autophagy was not considered as it was first described in the damage response by Lee et al., 2020.

We now demonstrate in new experiments that LAMP2 degradation is mediated by microautophagy because LAMP2 degradation is inhibited by STK38 knockdown (EV5 A and B). Therefore, by showing that also ATXN3 KO affects LAMP2 degradation, we demonstrate that ATXN3 facilitates microautophagy of LAMP2.

4. STK38 has been shown to be required for lysosomal membrane repair mediated by the ESCRT machinery, an early response to lysosomal damage [PBID: 37987447]. In contrast, the authors demonstrate that ATXN3 is recruited to damaged lysosomes at a later stage and is not involved in ESCRT-mediated repair. Therefore, the proposed model in which ATXN3 promotes the degradation of a fraction of LAMP2 via turnover of its K48-K63-branched ubiquitin conjugates through STK38-mediated microautophagy appears inconsistent in both timing and mechanism. For these reasons, outlined in points #2, #3, and #4, the interpretation of Figure 6 remains unconvincing.

We disagree. In contrast to the referee's statement, the cited paper (Ogura et al., 2023, PMID: 37987447) demonstrates that STK38 localizes to lysosomes and acts in microautophagy 3 h after LLOMe washout. This perfectly aligns with the timing of ATXN3 function which we show peaks at 2-4 hrs after LLOMe washout. In addition, we now show directly that STK38 (like ATXN3) facilitates LAMP2 degradation after damage. This is consistent with our conclusion that both STK38 and ATXN3 mediate microautophagy. Ogura and colleagues used the term "repair" to describe the function of STK38, but they did not assay the initial lysosomal membrane resealing and reacidification that was described as "repair" by the Hanson group and others, and that the referee likely refers to. Therefore, STK38, like ATXN3, acts in the hours following initial membrane sealing.

Prof. Hemmo Meyer
Universität Duisburg-Essen
Faculty of Biology
Universitätsstr. 2
Essen 45141
Germany

11th Jul 2025

Re: EMBOJ-2024-118858R1
ATXN3 regulates lysosome regeneration after damage by targeting K48-K63-branched ubiquitin chains

Dear Hemmo,

Thank you for submitting your re-revised manuscript for our consideration. I have now had a chance to look through it and to assess your responses to the comments raised by the original reviewers, and I am happy to inform you that there are no further objections towards publication in The EMBO Journal.

With kind regards,

Hartmut
